# Cost estimation for the monitoring instrumentalization of Landslide Early Warning Systems

Marta Sapena[1], Moritz Gamperl[2], Marlene Kühnl[1,3], Carolina Garcia-Londoño[4,5], John Singer[6], Hannes Taubenböck[1,7]

[1]German Aerospace Center (DLR), German Remote Sensing Data Center (DFD), Münchnerstrasse 20, 82234 Weßling, Germany
[2]Technical University of Munich, Chair of Engineering Geology, Arcisstrasse 21, 80333 München, Germany
[3]Company for Remote Sensing and Environmental Research (SLU), Kohlsteinerstrasse 5, 81243 München, Germany
[4]Leibniz Universität Hannover, Institut für Landschaftsarchitektur, Herrenhäuserstrasse 2a, 30419 Hannover, Germany
[5]Geological Society of Colombia, Colombia
[6]AlpGeorisk, Donauwörth, Germany
[7]Institute for Geography and Geology, Julius-Maximilians-Universität Würzburg, 97074 Würzburg, Germany

*Correspondence to*: M. Sapena (marta.sapena-moll@dlr.de)

**Abstract.** Landslides are socio-natural hazards. In Colombia, for example, these are the most frequent hazards. The interplay of climate change and the mostly informal growth of cities in landslide-prone areas increases the associated risks. Landslide Early Warning Systems (LEWSs) are essential for disaster risk reduction, but the monitoring component is often based on expensive sensor systems. This study presents a data-driven approach to localize landslide-prone areas suitable for low-cost and easy-to-use LEWS instrumentalization, as well as to estimate the associated costs. The approach is exemplified in the landslide-prone city of Medellín, Colombia. A workflow that enables decision-makers to balance financial costs and the potential to protect exposed populations is introduced. To achieve this, city-level landslide susceptibility is mapped using data on hazard levels, landslide inventories, geological and topographic factors, and a random forest model. Then, the landslide susceptibility map is combined with a population density map to identify highly exposed areas. Subsequently, a cost function is defined to estimate the cost of LEWS-monitoring sensors at the selected sites, using lessons learned from a pilot LEWS in Bello Oriente, a neighbourhood in Medellín. This study estimates that LEWS monitoring sensors could be installed in several landslide-prone areas with a budget ranging from €5 to €41 per person (roughly COP 23,000 to 209,000), improving the resilience of over 190,000 exposed individuals, 81% of whom are located in precarious neighbourhoods; thus, the systems would particularly reduce the risks of a social group of very high vulnerability. The synopsis of all information allows to provide recommendations for stakeholders on where to proceed with LEWS instrumentalization. These are based on five different cost-effective scenarios. This approach enables decision-makers to prioritize LEWS deployment to protect exposed populations while balancing the financial costs, particularly for those in precarious neighbourhoods. Finally, the limitations, challenges, and opportunities for the successful implementation of a LEWS are discussed.

# 1 Introduction

The lives and livelihoods of billions of people around the world are disrupted by human-induced hazards, exacerbated by climate change. The Intergovernmental Panel on Climate Change (IPCC) of the United Nations has recently reported that climate change is causing more frequent and severe storms, floods, droughts, wildfires, and other extreme weather events (IPCC, 2022). The climate crisis has global implications and poses several challenges to governments, societies, and science (Marchezini et al., 2018), with some geographic regions being more affected than others. For example, Colombia is one of the most landslide-prone countries in the world. The majority of its population lives in areas that are prone to landslide hazards (Ruiz Peña et al., 2017). In Colombia, the increasing frequency of intense and persistent rainfall, coupled with unplanned urban growth in areas prone to landslides driven by limited land availability, significantly increases the likelihood of disasters, particularly impacting the most vulnerable populations (World Bank, 2012).

Due to the occurrence of a disaster, or to mitigate the effects of an imminent natural hazard, people are occasionally compelled to evacuate their places of residence. Displacement severely disrupts people's lives, rising unemployment, interrupting education, and hindering access to essential services, ultimately leading to increased vulnerability and impoverishment. Implementing preparedness measures is crucial in mitigating risks associated with displacement. These measures enhance risk awareness among individuals at risk of displacement, empowering them to make informed decisions and comply with the warnings (UNDRR, 2021). In fact, one of the seven global targets of The Sendai Framework for Disaster Risk Reduction 2015–2030 is to "*Substantially increase the availability of and access to multi-hazard early warning systems and disaster risk information and assessments to people by 2030*" (UN, 2015).

Landslide Early Warning Systems (LEWSs) play a crucial role in reducing the risk associated with landslides. LEWSs provide timely information of slowly changing slope stability or acutely dangerous situations, enabling proactive measures, enhancing public awareness and education, and facilitating better planning and decision-making. Additionally, LEWSs collect valuable data that can be utilized for scientific research, monitoring, and analysing landslide behaviour. Thus, in turn, the understanding of landslides, their triggers, and their impacts can be understood better, leading to the development of improved predictive models and more effective LEWSs (Guzzetti et al., 2020; WMO, 2018; Segoni et al., 2023). LEWSs have the potential to minimize the loss of lives and mitigate the economic and social impacts of disasters. They provide significant economic benefits by reducing damage and loss, facilitating cost-effective planning and response, preserving economic activities, and saving costs in emergency response operations (Rogers and Tsirkunov, 2011; Grasso, 2014). Therefore, they offer a viable alternative to relocating exposed populations, especially since relocation is in most regions economically unviable and relocation often faces strong opposition from residents (Werthmann and Echeverri, 2013). These economic advantages make investing in LEWSs a prudent choice. However, it is important to note that the cost of implementing LEWSs can vary significantly depending on factors such as the size and complexity of the monitored area, the technology and infrastructure employed, and the level of system sophistication. Furthermore, beyond the technical implementation and maintenance, the

effectiveness of a LEWS relies on actively involving at-risk individuals, improving education and awareness of risks,

efficiently disseminating messages and warnings, and ensuring preparedness (WMO, 2018).

In general, however, LEWSs offer promising potential when well-integrated and properly managed. However, these systems also face several challenges, shortcomings, and untapped potential: The monitoring components of LEWSs often rely on expensive high-end sensor systems, such as multi-phase GNSS, GB-SAR, and tacheometry. These systems require highly trained personnel for operation, and are specifically tailored to the local situation, making their transfer to other regions or

countries difficult. Nevertheless, there are also low-cost and easy-to-use sensor systems available, such as MEMS tilt inclinometers, acceleration sensors, continuous shear monitors, and piezometers. By installing these geosensors in a local network, valuable information about the surface and subsurface processes on landslide-prone slopes can be obtained (Thuro et al., 2014; Uchimura et al., 2015; Singer et al., 2021). When combined with data analysis and numerical landslide process models (Huggel et al., 2010; Thiebes et al., 2014), these sensor systems have the potential to enhance the quality, reliability,

and usability of hazard warnings, and to reduce the need for extensive manual work. An example is the low-cost subsurface monitoring system implemented in the Alps, specifically at Sudelfeld in Bayrischzell, Germany, which operated from 2008-2014. This system included cost-efficient ground deformation measures, groundwater level and precipitation monitoring (Singer and Thuro, 2006; Thuro et al., 2010). In recent decades, advancements in technology and affordability of slope monitoring have improved, enabling more widespread applications, including low-income countries. However, continuous

monitoring is still limited to only a few slopes worldwide, with some site-specific LEWS in Latin America (Guzzetti et al., 2020). The dissemination of experiences, challenges, and limitations associated with LEWSs is not a priority and is therefore rarely done, particularly in English (e.g., Reinoso Jerez, 2013; Departamento del Quindío, 2018; Castro Bonilla, 2021).

In general, there are estimates available regarding the economic benefits of Early Warning Systems (EWSs), particularly in European countries, the United States, and Japan (Hallegatte, 2012). For instance, in Europe, the annual benefits of EWSs are

estimated to range between 470 million and 2.8 billion Euros. Similarly, it has been projected that low-income countries could experience potential benefits with a cost-benefit ratio ranging from 4 to 35, along with associated co-benefits, provided similar EWSs were available (Hallegatte, 2012). However, estimating the costs and benefits of implementing local LEWSs remains challenging due to various factors influencing the overall expenses. The implementation of a comprehensive LEWS requires substantial investments, including but not limited to equipment costs, infrastructure development, ongoing maintenance,

personnel expenses, and social integration. Recognizing this knowledge gap, this study aims to contribute to the literature by providing cost estimations for the instrumentalization of a low-cost, local, and site-specific LEWSs. To the best of the authors' knowledge, the estimated cost of the monitoring instrumentalization of local and site-specific LEWSs is largely unknown or unpublished, despite being highly relevant for policy-makers involved in disaster risk reduction.

The purpose of a LEWS is to reduce risk and improve preparedness for hazards in specific locations. Thus, it is imperative to

identify landslide-prone areas, the location of people and assets in exposed locations, and their vulnerabilities. In this context,

Earth Observation (EO) plays an important role in early warning, mapping and monitoring natural hazards (Grasso, 2014; Casagli et al., 2017). EO can be used, for example, to identify populations exposed and vulnerable to different natural hazards (Taubenböck et al., 2008; Geiß et al., 2017; Müller et al., 2020; Kühnl et al., 2022), , as well as to estimate landslide susceptibility at different regions and scales (Cantarino et al., 2019; Palacio Cordoba et al., 2020; Ospina-Gutiérrez and Aristizábal, 2021).

Landslide susceptibility modeling has witnessed an increase in popularity due to the advancements in remote sensing, and machine learning models. Traditional knowledge-driven methods, such as the multicriteria analytical hierarchy process (AHP) developed by Saaty (1980), rely on weights assigned to several landslide-influencing factors. Thus, the result depends on the experience of the user and the potential to identify factors that are important for a special case (Günther et al., 2014; Skilodimou et al., 2019). In contrast, data-driven methods rely on reference data (e.g., landslide inventories) and conditioning factors, (i.e., factors influencing landslide risks), which are used to identify their interconnected relationships and predict landslide susceptibility based on statistical models. Data-driven approaches have demonstrated significant potential in effectively mapping areas prone to landslides, particularly in situations where the availability of comprehensive geotechnical data required for physically-based methods are lacking. Some of the most common data-driven methods include Random forest (Taalab et al., 2018; Calderón-Guevara et al., 2022; Ado et al., 2022; Abu El-Magd et al., 2021), Logistic regression (Ado et al., 2022; Azarafza et al., 2021), Convolutional and Artificial Neuronal Networks (Nikoobakht et al., 2022; Calderón-Guevara et al., 2022; Ado et al., 2022; Azarafza et al., 2021), Boosted Regression Trees (Calderón-Guevara et al., 2022; Pourghasemi et al., 2021), Weight of Evidence (Calderón-Guevara et al., 2022), Supported Vector Machine (Nikoobakht et al., 2022; Ado et al., 2022; Azarafza et al., 2021), K-Nearest Neighbor (Nikoobakht et al., 2022; Abu El-Magd et al., 2021), Naïve Bayes (Abu El-Magd et al., 2021; Azarafza et al., 2021), and Linear discriminant analysis (Eiras et al., 2021; Pourghasemi et al., 2021). Previous studies have compared the performance of AHP and statistical methods, and the latter was found to perform better (Erener et al., 2016; Ali et al., 2021; Vojtek et al., 2021). Nevertheless, currently, there is no definitive data-driven method established as the optimal choice for empirical landslide susceptibility modeling. In recent literature, various methods have been employed, compared, and their accuracies and suitability have shown regional variations. In this study, the Random Forest method is implemented due to its demonstrated high accuracy in Colombia (Calderón-Guevara et al., 2022). Additionally, this method offers the advantage of being non-parametric, allowing for the inclusion of not-normally distributed influencing factors (Breiman, 2001).

In this context, the landslide-prone city of Medellín in Colombia is notable for its comprehensive landslide inventories (which are crucial resources for collecting information related to landslides occurrences, for deriving empirical knowledge, and for creating landslide susceptibility maps), and its implementation of both, a city-wide EWS and a local-scale LEWS. On the one hand, the Aburrá valley and the city of Medellín implemented an EWS (*Sistema de Alerta Temprana de Medellín y el Valle de Aburrá*, SIATA), that monitors real-time hydrological, meteorological, seismic, and geotechnical variables, to forecast natural and anthropic phenomena and to strengthen risk management in the territory (SIATA, 2023). On the other hand, a unique local,

site-specific, and low-cost participatory LEWS was implemented in a local community known as Bello Oriente by a research project called Inform@Risk (Werthmann et al., 2023). Bello Oriente is a settlement that was originally built informally on one of the eastern slopes of the city in an area exposed to landslide hazard. The LEWS includes a wireless network of sensors that is based on Internet of Things (IoT) technologies. It monitors movements in the subsurface and their effects on the built infrastructure (e.g., tilting, opening of cracks), groundwater levels, and other parameters that, in combination with weather variables and forecasts, are used to inform people about the level of risk. With this, the system aims to provide exposed people enough time to react in the case of an event and to improve their risk awareness and resilience.

For the above-mentioned reasons, Medellín is selected as the study area. Using lessons learnt from the LEWS prototype installation in Bello Oriente in the frame of the Inform@Risk project, and assuming that sufficient financial resources are not available on an ad hoc basis for citywide instrumentation with LEWSs, a cost function is developed which allows to weigh how much money, on which location(s), and how many people can possibly benefit from a LEWS in landslide-prone areas. The ultimate goal is to assist decision-makers in their prioritization efforts, by providing a comprehensive assessment that facilitates the strategic implementation of LEWSs, ensuring maximum impact and cost-efficiency, by offering valuable recommendations based on different cost-effectiveness scenarios. Therefore, the specific objectives of this study are: (1) to identify highly exposed landslide-prone areas that are suitable for the implementation of LEWSs with automatic monitoring sensors; (2) to determine a cost function based on topographic and socioeconomic parameters for the implementation of a LEWS; and (3) to provide suggestions for decision-makers on where to start with the implementation of new LEWSs based on cost-effectiveness and prioritized areas.

## 2 Material and methods

In this section, the developed workflow for a city-wide cost-effectiveness analysis for LEWSs is introduced. First, the study area and datasets are presented. Subsequently, the process of deriving a landslide susceptibly map is explained, along with the preselection of exposed sites where the installation of a LEWS is preferable. Then, the calculation of physical, social, demographic, and infrastructure parameters is conducted for the preselected sites within the pool of possible LEWS installations. This analysis supports the identification of suitable sites for the LEWS installation. Lastly, the cost function is presented, which aims to prioritize the installation of LEWS based on cost-effectiveness (Fig. 1).

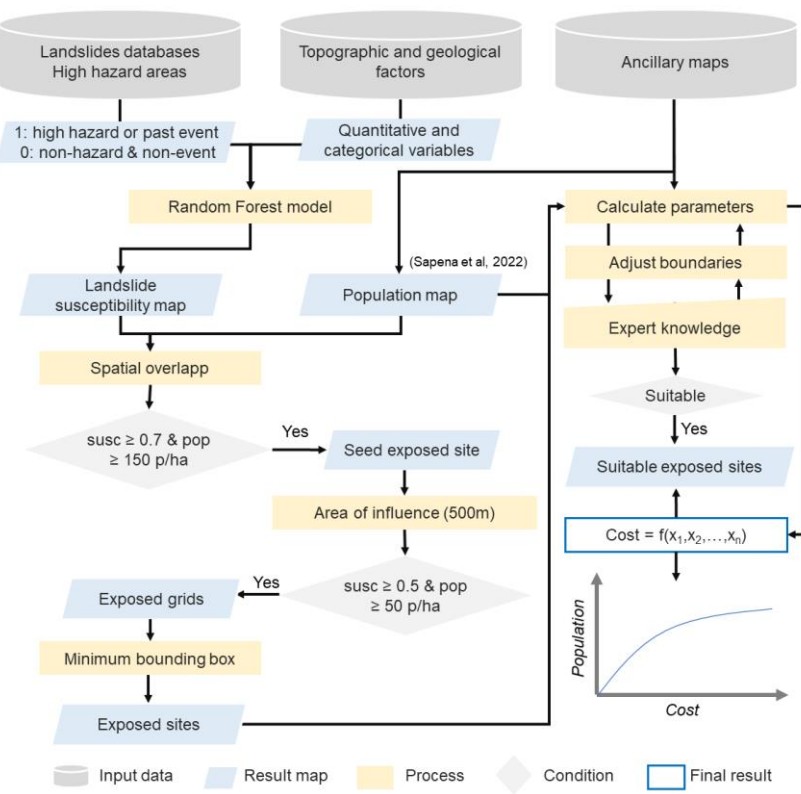

**Figure 1.** Workflow of the study.

## 2.1 Study area

This study takes place in Medellín, Colombia, a highly populated city situated in the Aburrá valley. The city is characterized by a steep topography, which poses a significant landslide risk, particularly on the eastern slopes. These slopes are predominantly composed of heavily fractured dunite rock, which undergoes deep and intense weathering in the region's tropical

conditions due to its high iron content (Thuro et al., 2020). The dunite is overlaid by colluvium material, previously displaced by landslide processes, forming a block-in matrix structure with varying thicknesses ranging from 5 meters to over 30 meters (Breuninger et al., 2021). This colluvium reveals that substantial proportions of the slope have been affected by landslides in the past, and the steep topography of the hills makes them susceptible to future landslides.

In Medellín an increasing portion of the population is exposed to landslide risks (Kühnl et al., 2022). The city showcases a

165 diverse range of urban structural configurations and socioeconomic levels, encompassing high-rise buildings in the business district and wealthier neighbourhoods, as well as light-weight and low-rise buildings in precarious neighbourhoods. The latter are primarily located in landslide-prone regions (Kühnl et al., 2021). These factors contribute to significant inequalities in risk exposure for various social groups.

## 2.2 Landslide inventories and hazard map

The extensive history of landslides in Medellín has led to the establishment of multiple databases that document landslide events, causalities, and damage to buildings and infrastructure. Specifically, there are three main databases where individuals report landslides: SIMMA, DesInventar and DAGRD.

SIMMA (*Sistema de Información de Movimientos en Masa*), managed by the Colombian Geological Survey, is a national landslide inventory (SIMMA, 2022). For Medellín, it provides precise coordinates for 13 landslides events between 1985 and 175 2013.

DesInventar is an international Disaster Information Management System that helps to analyse disaster trends and impacts (DesInventar, 2022). The 'Medellín Área Metropolitana' database, managed by the Universidad Nacional de Colombia, contains information from 1880 to 2022, including 21 landslides with precise coordinates between 2018 to 2022.

DAGRD (*Departamento Administrativo de Gestión del Riesgo de Desastres*) registers occurred or foreseen landslides based 180 on emergency calls from citizens, which are subsequently evaluated by DAGRD technicians. The department provided data from 2004 to 2019, encompassing over 30,200 reports of potential mass movements with their coordinates (DAGRD, 2018). It is important to note that the vast majority of these reports are concentrated in the urban areas, with limited coverage of events in rural regions.

In addition to landslide inventories, remote sensing data and techniques have proven effective for detecting and mapping 185 landslides (Guzzetti et al., 2012), especially when inventories lack precise spatial accuracy. In this paper, the information from the three inventories was combined with landslide locations identified through remote sensing in the urban-rural border as well as rural areas. This involves utilizing multi-temporal imagery from Landsat-7, Landsat-8 and Sentinel-2 satellites, and applying change detection methods to vegetation indices in areas where landslides are indicated by news articles. It is assumed that significant changes of these vegetation indices in areas of steep slopes are indicative of variations in the soil surface or 190 vegetation removal. These areas can then be attributed to landslides (Mondini et al., 2011). In total, 8 landslides were identified between 2008 and 2019.

Additionally, the latest Master Plan of the city of Medellín from 2014 (*Plan de ordenamiento territorial*, POT 2014) incorporates a zoning system for landslide hazards (Alcaldía de Medellín, 2014a). The hazard map results from a comprehensive analysis that incorporates various sources of information. These include the hazard map from the POT 2006, a 195 probabilistic hazard map developed by the Universidad Nacional, the DAGRD landslide inventory until 2014, as well as heuristic processes, fieldwork and the expertise of technicians from DAGRD and the Administrative Department of Planning (*Departamento Administrativo de Planeación*, DAP). To model landslide susceptibility, the mass movements inventories and the hazard map from the POT 2014 are utilized as reference data. Given the absence of specific details concerning the type of mass movements and considering that the majority of mass movements in Medellín are landslides, it is assumed that all

available reports correspond to landslides. These datasets form the foundation for training and evaluating the model for the identification of possible areas prone to landslides, as elaborated further in Sect. 2.4.

## 2.3 Factors influencing landslide risk

For modelling landslide susceptibility, the proposed methodology incorporates a range of factors influencing landslide risks including topographic, geological, and precipitation data. In addition, to support the search for suitable locations for the
implementation of LEWSs, socio-demographic factors are also considered. The database consists of the official cartography from open data platforms of the city of Medellín and the metropolitan area of the Aburrá valley such as 'GeoMedellín' and 'Datos Abiertos del AMVA' (Alcaldía de Medellín, 2023; AMVA, 2023), precipitation data from SIATA (SIATA, 2023), OpenStreetMap data (downloaded in 2022, openstreetmap.org), a high resolution population map that provides estimates of the number of people per building and a grid of 100 meters from Sapena et al. (2022), and a map of precarious settlements
from Kühnl et al. (2021).

Various topographic factors, are derived from contour lines to analyse the terrain. Contour lines at different scales (1:2,000 for urban areas and 1:5,000 for rural areas) are utilized to generate a Triangular Irregular Network (TIN) surface. By interpolating the altitude data from the TIN, a Digital Elevation Model (DEM) with a spatial resolution of 5 meters is generated. This DEM is used for deriving several topography-related factors, including slope, aspect (which indicates the downhill direction of the
slope), and curvature (which represents the shape or curvature of the slope) (Fig. 2). Additionally, the DEM is used to model water flows, resulting in the extraction of the stream network, stream order, landslide travel paths, and angle of reach of landslides (also known as the 'fahrböschung angle' or $\alpha$). The angle of reach provides insight into the potential mobility of a landslide (Hungr et al., 2005). Regarding the stream network, there is no universally agreed upon flow accumulation threshold for determining streams due to its dependence on various factors, including desired stream density, data scale, resolution, and
landscape attributes. Nonetheless, it is common practice to employ threshold values within a range of approximately 100 to 1000 pixels or 0.05 to 5 km² drainage area (Reddy et al., 2018; Tarboton et al., 1991). In this study, multiple values within this range are evaluated, and determined that a threshold of 500 contributing pixels (equivalent to a minimum stream inflow of 12,500 m²) achieved the closest correspondence with the official drainage system map from the POT of Medellín. Furthermore, a visual examination of the result reveals a satisfactory representation of the majority of streams. The Strahler method
(Tarboton et al., 1991) is used to classify the streams into numerical orders, distinguishing between main streams classified as major tributaries (ranging from order 7 to 5), and other streams categorized as outermost tributaries (ranging from order 4 to 1). Two Euclidean distance maps are then generated to calculate the distance from a given pixel to both, the main streams and the other streams (Fig. 2). Subsequently, the travel path and angle of reach of a landslide are used to support identifying exposed sites. This is done by identifying unpopulated areas where landslides are likely to occur and where their runout can
extend into the inhabited area.

For the geology, geomorphology, and geotechnics of the study area, data are gathered from 'Datos Abiertos del AMVA'. Three maps containing categorical information on geological units, geomorphologic units, and geotechnical zoning are used. The surface geology plays a crucial factor as it informs about the physical composition of materials, their properties, and mechanical strength characteristics. Geomorphology offers valuable information on the landscape, i.e. the stability, slope, and shape. Additionally, geotechnics provide details about soil types (Universidad Nacional de Colombia, 2009) (Fig. 2).

Precipitation plays a crucial role in triggering landslides, particularly in mountainous areas. In the study area, the average yearly precipitation exhibits significant variations due to local luv and lee effects. Thus, precipitation data obtained from the meteorological stations of SIATA are used. Precipitation measurements from the 215 available stations in the Aburrá valley for 2021 are acquired. Using these data points, interpolation techniques are employed to generate a continuous map of rainfall accumulation with a spatial resolution of 5 meters. To achieve this, the Ordinary Kriging Optimized Smoothed (OKOS) method is employed, accompanied by cross-validation. For this process, 70% of the stations are selected for training and the remaining 30% for testing. The evaluation provides a Root Mean Square Error (RMSE) value of 506 mm/year, which represents the discrepancy between the predicted and observed yearly precipitation accumulation. In relation to the mean value of precipitation across all stations, i.e. 1,425 mm/year, the normalized RMSE, indicating the relative error, is approximately $\pm$ 35%. This error is in line with existing literature, which commonly reports normalized RMSE values ranging from 30-35% (Bostan et al., 2012) and 17-29% (Antal et al., 2021) (Fig. 2).

When it comes to decisive factors for assessing the suitability and prioritization of potential locations for implementing a LEWS, socio-demographic and infrastructure variables are used. To begin with, the population map is used to identify sites with both, high population density and a high susceptibility to landslides. These areas allow to quantity the exposed population. However, from a socio-economic perspective the exposed populations at these sites does not have equal vulnerabilities, i.e., for example, the financial resources to recover from a disastrous event. In consequence, the calculated population of these sites is combined with information on the socio-economic precariousness of the residents. The exposed population living in precarious settlements is referred hereinafter as highly vulnerable. This comprehensive analysis enables to prioritize exposed areas that exhibit a higher vulnerability to disastrous events. In addition, data on the road infrastructure is employed, which is a relevant factor in determining the placement of sensors within the LEWS. The official road network cartography from 'GeoMedellín' is supplemented with additional road data from OpenStreetMap (Fig. 2).

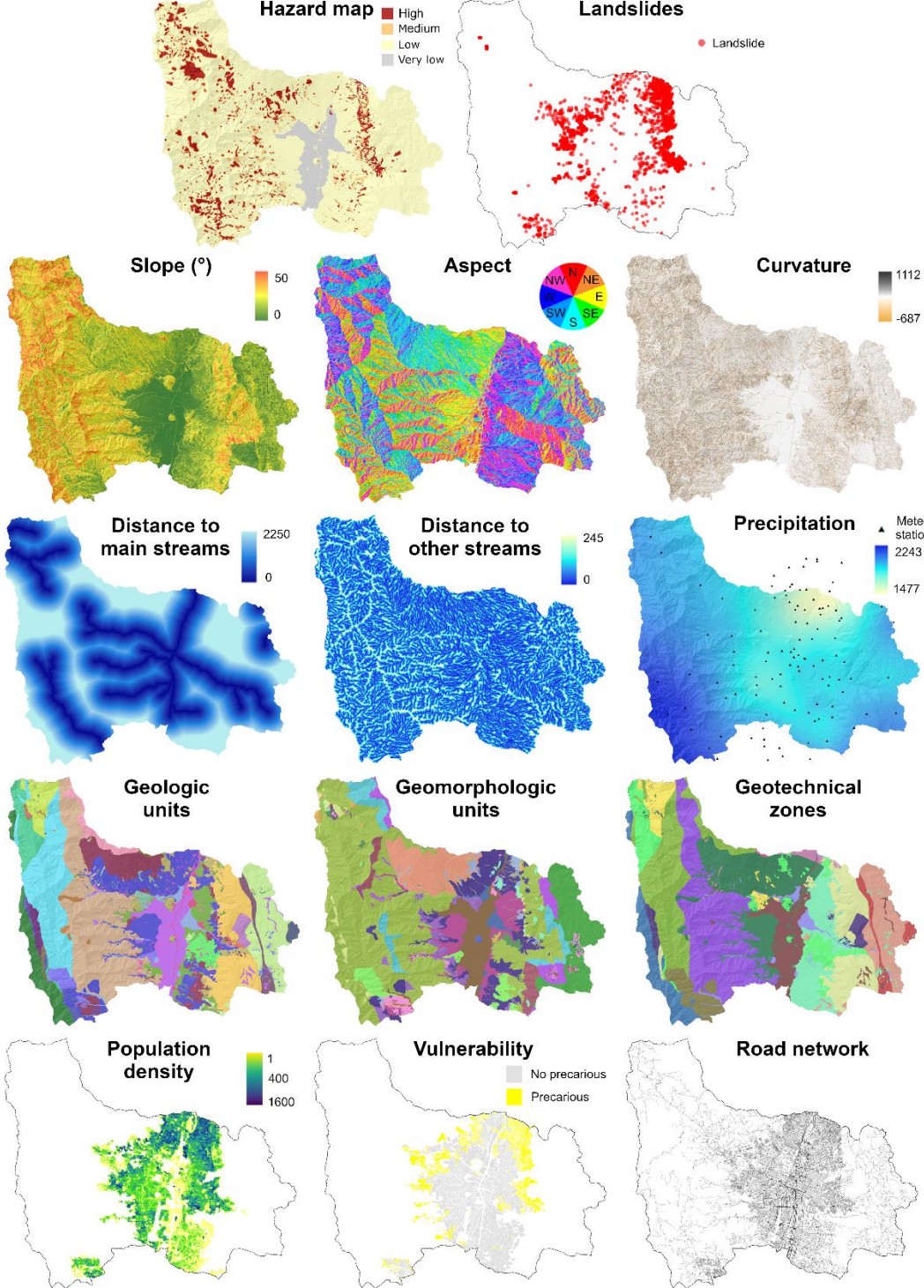

**Figure 2.** Dependent variables in the model (hazard map and landslide events), independent variables or conditioning factors (topographic, geological, and precipitation), and ancillary data (socio-demographic and infrastructure).

## 2.4 Mapping landslide prone areas

In this study, baseline data on recent landslide events and an official hazard map are employed to train a Random Forest statistical model. The aim of this model is to predict the probability of landslide events, also referred as landslide susceptibility, by considering multiple conditioning factors. Therefore, the landslide hazard map, which incorporates local expert knowledge, is complemented using additional landslides events from the compiled inventories that were not originally included in the hazard map (i.e., all landslides from Sect. 2.2 starting from 2015 onwards). Both datasets are considered as reference data in the model. Consequently, a set of sampling points is created using these reference data: within high-hazard zones (5,000 points), recorded landslide events (2,800 points), and non-hazardous areas including medium, low or very low hazard zones (8,000 points). Each point is assigned a numeric attribute representing the dependent variable: high-hazard (1) and non-hazard (0). Additionally, the values of all conditioning factors, explained in Sect. 2.3, are also assigned to each point and are used as independent variables in the model. The sampling set is then divided into training data (70%) and testing data (30%). The non-parametric Random Forest (RF) algorithm (Breiman, 2001) is employed, using 100 trees, 3 variables at each split, and 5 terminal nodes. The susceptibility map is generated using the training data, providing the probability of a pixel being classified as high-hazard (1), calculated based on the proportion of votes across the ensemble of trees. The testing data, which was not used for training, is used to evaluate the model. Ultimately, the accuracy statistics are measured using the remaining testing set.

## 2.5 Selection of exposed areas

A zero-susceptibility level is assigned to areas with slopes equal to or lower than 10° in the resulting landslide susceptibility map. This is in accordance with the assessment by Alcaldía de Medellín (2014b) as these slopes are considered non-hazardous in Medellín. Next, to identify the exposed areas, the average susceptibility per 100-meter grid cell is calculated and combined with the population density per grid. This analysis focuses on the urbanized area, which contains the most exposed elements, and where population data at a fine grain level is available.

The identification of exposed sites follows an iterative process, as depicted in Fig. 1. First, seeds of exposed sites are identified based on high susceptibility (≥ 0.7) and high population density (≥150 people/ha). If two or more seeds are contiguous or within 100 meters of each other, the centroid is used as the seed location. Second, these seeds serve as starting points to search within a 500-meter area of influence, which is set to match the size of the reference LEWS prototype in Bello Oriente. Pixels with medium to high susceptibility (≥ 0.5) and medium to high population density (≥50 people/ha) within this area are considered. This process identifies all susceptible and populated pixels surrounding the seeds. Third, a minimum bounding box is calculated for each cluster of pixels to automatically define a preselected exposed site. Fourth, the shape of the preselected sites is adjusted based on factors such as the urban structure, topography, travel path and angle of reach of a landslide. Finally, several parameters for the preselected sites are calculated, including area, mean susceptibility, total population, population density, vulnerable population, built-up and road density, number of buildings, mean slope, main

orientation of the slope, and open areas for the preselected sites. These parameters are then used by experts to inspect and select the suitable exposed sites based on the requirements of the LEWS and previous experience in Bello Oriente.

## 2.6 Developing a cost function

To estimate the costs of monitoring instrumentalization for LEWSs in suitable sites, the existing LEWS prototype in Bello Oriente serves as a reference. It provides information on the required manpower (working hours for the sensor system construction and installation), node density, node types, and individual node costs. The Bello Oriente LEWS prototype covers a 39-hectare area with a total population of 4,600 residents, including 1,800 individuals residing in high-hazard landslide zones. The Bello Oriente LEWS prototype utilizes an innovative wireless geosensor network based on IoT technologies, such as

LoRa® wireless data communication and MEMS sensors. It incorporates 1,100 meters of Continuous Shear Monitor measurement cable and extensometers (CSM-EXT) (Thuro et al., 2010) to monitor subsurface movements and near-surface groundwater levels, forming the foundation for generating warnings. Given the unpredictable nature of future landslides based on geological investigations, extensive coverage with high spatial and temporal observation density is essential. The wireless geosensor network in Bello Oriente consists of 115 nodes, of which 45 monitor subsurface deformation and groundwater levels

(Subsurface Nodes), while 70 nodes detect movement in existing built infrastructure (Infrastructure Nodes). The spacing of nodes is adjusted according to the level of landslide risk, with high-risk areas having on average 8 nodes per hectare and areas with no risk having no nodes. The nodes are open-source and can be replicated using published PCB schematics, 3D printing models, and building instructions. Further details about the measurement system can be found in Gamperl et al., (2021) and Singer et al., (2021), as well as on the Inform@Risk Wiki ([www.informatrisk.com](www.informatrisk.com)).

It is important to note that the current cost estimation only considers the implementation of the wireless geosensor network and does not encompass other factors. Costs related to risk evaluation, social interventions, dissemination, continuous maintenance, and social work are not included in the analysis. The CSM-EXT measurement system was excluded due to its complexity and high installation costs, particularly in densely populated areas. Consequently, its widespread implementation as an alternative to the wireless sensor network might not be viable or cost-effective without conducting a detailed on-site

survey. The costs associated with social work are highly site-specific and depend on various factors. These factors include whether the municipality conducts a risk assessment on the site, the community's acceptance of the LEWS installation, and the involvement of NGOs working with the community. Furthermore, the extent and nature of social work can vary significantly based on the risk awareness and social structure of the community. Drawing from the experience of the Inform@Risk project, the initial social implementation costs of the LEWS, which mainly involves social workers

accompanying and explaining the installation process, producing and distributing information materials, and conducting training workshops and emergency drills, are expected to be at least comparable to the cost of implementing the technical system.

Next, the variables included in the cost function are explained. The cost of the sensor system elements was determined based on the required working time for production and installation, the 3D printing time, and the material costs such as electronics, sensors, cables, connectors, and accessories. For the working time, an hourly rate of €15 was used, which corresponds to the approximate hourly cost (inclusive of all insurances and benefits) of a geotechnician in Colombia (as of 2022). The cost of 3D printing time was estimated at €3 per hour, considering the filament cost, power consumption, maintenance, and the overall investment cost for the 3D printer spread over the lifespan of 10,000 operating hours. Material costs encompass all the components needed to construct and install the system. The costs are calculated without including VAT. For a detailed list of the required materials and step-by-step instructions for constructing and installing the sensor system, please refer to the Inform@Risk Wiki.

Regarding the sensor system design, a node density of 8 nodes per hectare is established for areas with the highest susceptibility level of 1, resulting in a dense grid of nodes. This density is scaled down based on the susceptibility (i.e., a susceptibility of 0.5 corresponds to 4 nodes per hectare). The built-up density determines the distribution of node types. In highly urbanized areas, more infrastructure nodes are preferred, while in less urbanized areas, more subsurface nodes are considered. Therefore, the node density is multiplied by the total area to determine the necessary number of nodes per site. Subsequently, this number is multiplied by the built-up density to calculate the amount of infrastructure nodes, with the remaining nodes being assigned as subsurface nodes. As for the gateways, it is assumed that at least one gateway per 25 hectares is necessary, although having at least two gateways for redundancy is suggested to ensure a backup in case of failure. This is a conservative assumption, as typically not as many gateways will be required. Previous tests conducted in the city of Medellín indicated that using 2 to 4 gateways in the city centre could provide sufficient transmission reach to cover the entire eastern slope of the city. The cost for the three different sensor systems is presented in Table 1.

**Table 1.** Cost for the different monitoring sensor systems.

| System | 3D Printing time and cost (€3/h) | Working time and cost (€15/h) | Material cost | Total cost |
|---|---|---|---|---|
| Infrastructure Node | 4.4h €13.2 | 1.5h – 1.75h €22.5 | €215 | €250 |
| Subsurface Node | 40.7h €122.1 | 3.75h €56.25 | €355 | €535 |
| Gateway | - | 8h €120 | €2,100 | €2,220 |

The cost function is calculated following Eq. (1):

$$COST = \text{S} \times 8 \times \text{A} \times (B_{DENS} \times €250 + (1 - B_{DENS}) \times €535) + \text{G} \times €2{,}220, \tag{1}$$

where $\text{G} = \begin{cases} A \leq 25\ ha = 1 \\ A > 25\ ha\ \&\ A \leq 50\ ha = 2 \\ A > 50\ ha = 3 \end{cases}$

The $COST$ represents the estimation of monitoring instruments costs for a LEWS in a specific site. The variables considered in the cost function include the landslide susceptibility (S), the area of the site in hectares (A), and the built-up density within the site ($B_{DENS}$).

The cost function results are used to develop alternative cost-effectiveness scenarios to support decision-making for the implementation of new LEWSs. Five scenarios are evaluated, including prioritizing (1) the total cost of the LEWS, (2) the cost per person, (3) the total population (exposed and vulnerable), (4) the landslide susceptibility, and (5) a combination of the aforementioned scenarios (1-4). In scenarios (1) and (2), priority is given to the lowest costs, while in scenarios (3) and (4), priority is given to the highest population and susceptibility. In the combined scenario (5), the values are normalized using the

min-max scale, where 1 represents the highest priority and 0 the lowest. The normalized values are then mapped and plotted on a graph, providing decision-making support.

## 3 Results

### 3.1 Landslide susceptibility map, exposed and suitable sites for LEWSs

The mapping of landslide susceptibility achieved an overall accuracy of 75.26% (Fig. 3a). The sensitivity and specificity of

the generated map were 80% and 71%, respectively. Sensitivity represents the percentage of correctly predicted high-hazard (1) class instances, while specificity represents the percentage of correctly predicted non-hazard (0) class instances. These metrics indicate that the model tends to slightly overestimate landslide susceptibility. It is important to note that although the reference data are discrete, the susceptibility map is a probability map with continuous values ranging from 0 to 1 (a detail can be seen in Fig. 3b). Therefore, the accuracy is measured by considering probabilities equal or higher than 0.5 as high-hazard

(1) and probabilities lower than 0.5 as non-hazard (0), without considering the degree of susceptibility for the validation metrics.

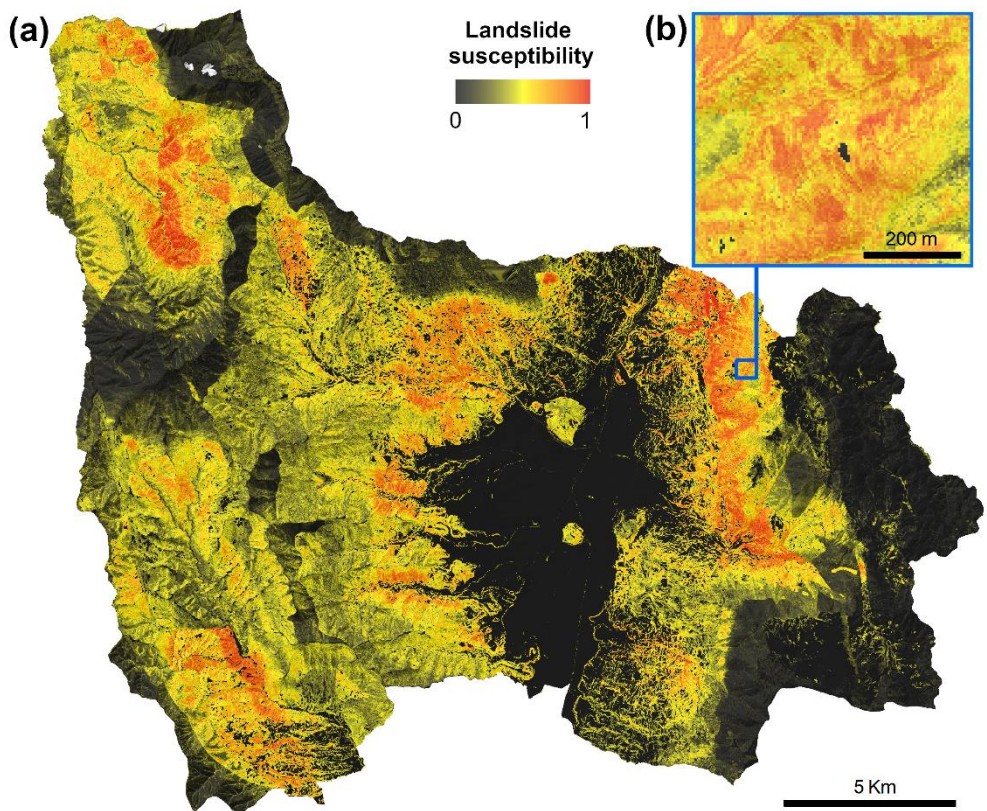

**Figure 3.** (a) Landslide susceptibility map (after filtering slopes equal or lower than 10°), and (b) detail in Bello Oriente neighbourhood.

To identify exposed sites suitable for potential LEWSs implementation considering the risk factors, the combination of
landslide susceptibility, and vulnerable and exposed elements was performed. A total of 44 seeds were identified (Fig. 4a and b), which served as reference points for identifying susceptible and populated areas surrounding them, thereby delineating the boundaries of the sites. Subsequently, socio-demographic and topographic factors were calculated and used to assess site suitability. Among the identified sites, a total of 16 sites were deemed unsuitable for node-based LEWS implementation based on expert recommendations (Fig. 4c). The primary reason for discarding these sites was their high building density, with
limited available open space for installing subsurface nodes and restricted monitoring capabilities of the LEWS to only infrastructure nodes. Additionally, some of the remining 28 pre-selected sites were subdivided into smaller areas or modified based on considerations such as topography, built-up density, and road network, in order to accurately define the boundaries of the suitable exposed sites (Fig. 4d).

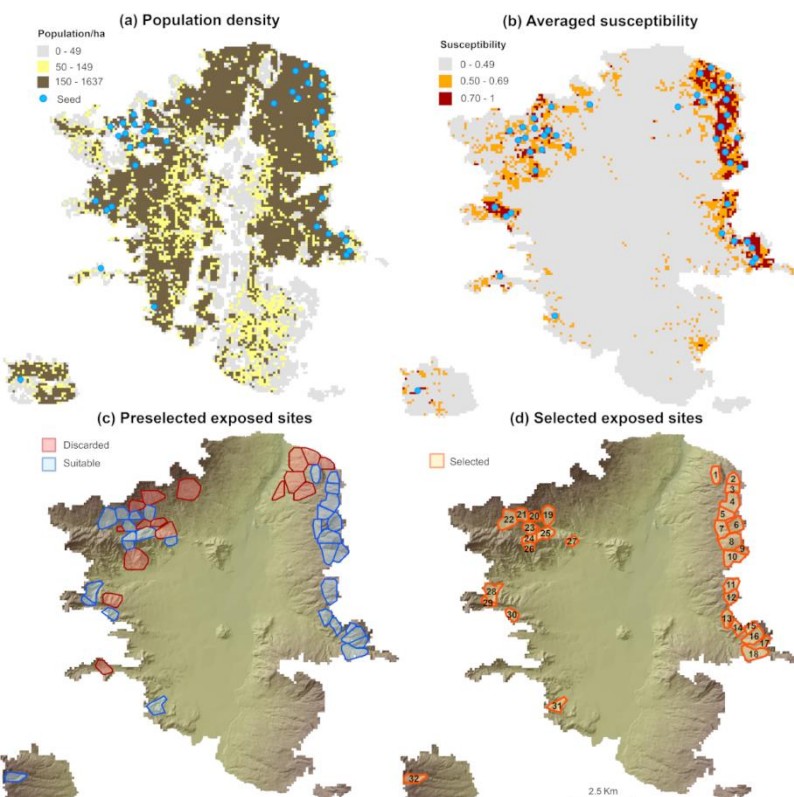

**Figure 4.** Identification of seeds based on (a) highly populated areas and (b) susceptible grids. (c) Automatic delineation of exposed sites by means of seeds' areas of influence and preselection of suitable sites. (d) Suitable exposed sites after the manual delineation of their boundaries.

The manual delineation process led to the identification of 32 preferred sites suitable for node-based LEWS installation (Fig. 4d). Socio-demographic and topographic factors used in the cost estimation function, were calculated for these definitive sites

(Table S1, supplementary material). These sites are primarily located in the north-eastern part of Medellín, with additional sites in the east and west, characterized by high population densities and high landslide hazards (see Fig. 2 and Fig. 4). On average, these sites cover an area of 27 hectares, with an average built-up density of 20%, and an average population density of 224 people per hectare (p/ha) (Table S1). The most densely populated site, located in *Área de expansion Pajarito* (site 21), has a population density of 512 p/ha and a low built-up density. This area corresponds to a city expansion zone with high-rise

buildings on the west side of the valley and no identified vulnerable population. On average, the sites encompass approximately 34% of open land, which is essential for installing subsurface nodes. In terms of slope, the selected sites have an average slope of 24°, ranging from a minimum of 15° in the west to a maximum of 35° in the east. As for landslide susceptibility, the average value is 0.68, with a minimum susceptibility of 0.51 and a maximum of 0.85 for the selected sites.

### 3.2 Cost estimation for the instrumentalization of LEWSs

Based on the generated landslide susceptibility map and the factors outlined in Table S1, the estimated costs for installing monitoring sensors in the selected sites for LEWSs are calculated using Eq. (1). In order to facilitate and guide decision-making regarding the next LEWS installation in the city, various cost-effectiveness scenarios are evaluated. These scenarios go beyond the monetary efficiency and consider additional priorities such as the number of exposed and vulnerable individuals, as well as the landslide susceptibility of each site. By considering these factors, a more comprehensive understanding of site

characteristics is obtained. Consequently, five priority scenarios are examined including prioritizing the total cost of LEWS, cost per person, total population exposed within the site (including vulnerable individuals), landslide susceptibility, and a combination of these factors.

Figure 5a provides information on the population that could potentially benefit from a LEWS in relation to the economic resources required in the combined scenario. It represents the overall costs of the system plotted against the total population

per site. The priority level is indicated by a greyscale gradient, with darker shades indicating higher priority. The cost of the systems ranged from €26,000 (≈ COP 132 Million, Colombian pesos, with a conversion rate of COP 5,040 per € at the time of writing) to €157,000 (≈ COP 789 Million). In terms of cost per person (p.p.), the estimated range was €5 to €41 p.p. (≈ COP 23,000 to 204,000). Therefore, if the goal is, for instance, to prioritize the most affordable system (i.e., focusing on LEWS cost), the LEWS in *El Corazón* (site 29) located on the western slopes of the city is the least expensive option (Fig. 5a).

However, it covers a smaller population compared to other sites with similar costs. In this sense, the LEWS in *El Pesebre* (site 27) is the second least expensive option for instrumentation. It covers more than double the exposed population compared to site 29, although it is important to note that most of the population is not classified as vulnerable.

To demonstrate the potential of the proposed cost function, a case scenario is simulated where the city of Medellín has a budget of COP 2,000,000,000 (≈ €397,000) allocated for the implementation of LEWSs. This simulation aims to illustrate the number

of LEWSs that could be installed within a specific budget, considering different cost-effectiveness scenarios. Based on these scenarios, several sites are suggested as potential starting points for implementing the systems, depending on different priorities. Figure 5b shows the locations of the sites where LEWSs could be installed using the allocated budget, according to the different scenarios (overall cost, cost p.p., exposed and vulnerable population, landslide susceptibility, and the combined scenarios) using values from Table S1. The colour assigned to each site represents the prioritization based on the corresponding

scenario. Sites that have more than one colour are prioritized in multiple scenarios. The accompanying table in Fig. 5b provides information on the total number of LEWSs that could be instrumentalized under each priority scenario, the total cost, average cost per person, the total number of exposed and vulnerable people, and the average susceptibility.

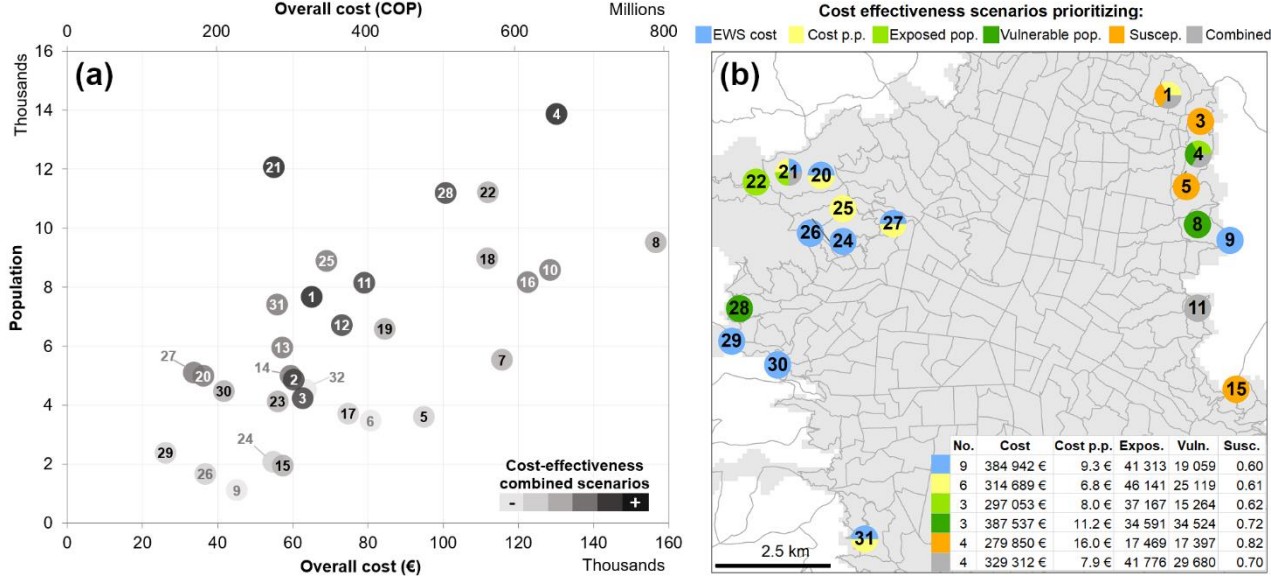

**Figure 5.** (a) Overall costs of the monitoring system installation of a site-specific LEWS versus the number of people for each site. The label number represents the site ID, while the grey tone represents the priority per site based on the cost-effectiveness of the combined scenario. Cost is given in Euros (€) and Colombian pesos (COP). (b) Based on the priority, the map shows the sites where LEWSs could be instrumentalized with an allocated budget of COP 2,000 Million. Values are summarized in a table with: the number of LEWSs, total cost, cost per person, people exposed, people vulnerable and susceptibility.

With the same available budget, it is feasible to instrumentalize nine LEWS if the objective is to minimize the total cost (shown in blue in Fig. 5b). These LEWSs are predominantly located on the western slopes and cover a population of approximately 41,000 individuals, including 19,000 who are considered highly vulnerable. The average landslide susceptibility in these areas is around 0.6. However, when considering all priorities, four LEWSs can be installed (represented in grey in Fig. 5b), at a lower average cost per person. These LEWSs are primarily located on the eastern slopes, with only one situated in the western part of the city. In this scenario, the number of exposed individuals covered by the LEWSs remains at 41,000, but the proportion of highly vulnerable individuals increases to nearly 30,000. Additionally, the sites exhibit higher susceptibility levels.

For instance, one site of great significance in several scenarios is located on the north-eastern slope of Medellín, between the south of *Carpinelo* and north of *Maria Cano-carambolas* (site 4). This site is home to almost 14,000 highly vulnerable individuals residing in precarious settlements, with a relatively high average landslide susceptibility of 0.74. The estimated cost p.p. at this site is €9.4 (COP 47,000), indicating that with an approximate budget of €130,000 (COP 656 Million), a substantial proportion of the exposed and highly vulnerable population can be covered. Likewise, the LEWS in *Santo Domingo el Savio 1* (site 1), located on the north-eastern slope, is the most effective in terms of cost per person, susceptibility, and combined priorities. However, if the objective is to cover the maximum number of exposed population, this site may not be the most suitable option. In that case, the LEWS in *Área de expansion de Pajarito* (site 21), situated on the western slopes, covers a greater number of exposed people, offers a more effective cost per person, and has a lower overall system cost

compared to site 1. However, its lower landslide susceptibility reduces the likelihood of a landslide occurrence. On the other hand, the LEWS in *El Corazón* (site 29) would be the most affordable option with a low cost per person, yet it exhibits the lowest susceptibility and covers fewer exposed individuals, which may impact its suitability. Similarly, the most expensive LEWS in *La Cruz* (site 8), on the north-eastern slope, has a reasonable cost per person, while the number of exposed individuals and the landslide susceptibility are significantly higher.

Figure S1 (supplementary material) illustrates the 32 sites in descending order of priority for the installation of site-specific LEWSs. The figure presents the cumulative value of the five priorities, where the exposed population is divided by vulnerability. By considering all priorities together, decision-makers can assess the combined value in Fig. S1, enabling them to identify sites that are highly prioritized across multiple cost-effectiveness scenarios simultaneously. This graphical representation assists in locating the optimal site for implementing the next LEWS, having into account various considerations
such as available funds (as depicted in Fig. 5b) or the size of the highly vulnerable population.

    Furthermore, Fig. 6 illustrates the cost function developed for prioritizing the installation of LEWSs based on the different cost-effective scenarios. If the city intends to implement LEWSs in all 32 proposed suitable sites, a total budget of €2.4 Million (≈COP 12,100 Million) would be required to cover the 200,000 exposed individuals. The trend lines in Fig. 6 depict the population covered by LEWSs based on the available funds and the priority scenario. Prioritizing the cost p.p. yields the most
efficient result in terms of budget utilization and population coverage. However, as observed in Fig. S1, this approach overlooks landslide susceptibility and the overall exposed population. In this regard, the combined scenario, which considers all relevant factors, demonstrates a similar trend, and is recommended for prioritizing the installation of new LEWSs. As an example, the priority of site 4 is displayed for all cost-effectiveness scenarios. By utilizing these scenarios, policy-makers can make conscious and informed decision regarding the installation of LEWSs, including the purpose and location.

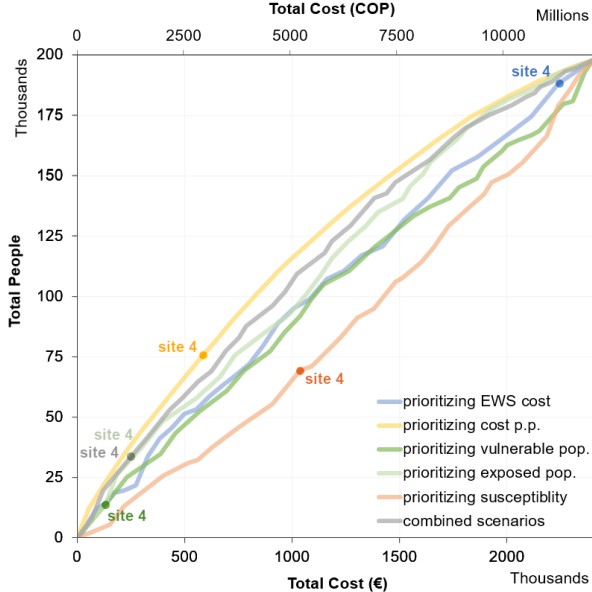

**Figure 6.** Cost function based on different cost-effectiveness scenarios and their combination. It shows the number of people that can be covered by the available funds, based on the preferred priority. Site 4 is highlighted to exemplify the particular priority within the cost-effectiveness scenario being considered.

## 4 Discussion

In recent years, there has been a significant increase in the utilization of data-driven methods and EO-derived data for mapping landslide susceptibility (e.g., Abu El-Magd et al., 2021; Ado et al., 2022; Azarafza et al., 2021; Calderón-Guevara et al., 2022; Eiras et al., 2021; Nikoobakht et al., 2022; Taalab et al., 2018) and for generating detailed population distribution maps (Sapena et al., 2022; Metzger et al., 2022). These advancements have enabled the identification of highly exposed areas prone to landslides in various regions worldwide (Garcia et al., 2016; Modugno et al., 2022; Kühnl et al., 2022). This study proposes a

comprehensive workflow that can be applied to identify exposed areas prone to landslides suitable for the implementation of low-cost and site-specific LEWSs. Furthermore, it adds to current literature as a cost estimation function for the instrumentalization of the LEWSs is developed, considering factors such as area, landslide susceptibility, and building density, allowing for the assessment and comparison of estimated costs across multiple sites. This integrated approach facilitates informed decision-making processes by prioritizing actions based on cost-effectiveness. One of the key contributions of this

study is the provision of an open and transparent cost estimation for LEWSs, serving as a valuable reference for other regions.

Through the application of the proposed workflow, more than thirty critical locations characterized by high exposure, high vulnerability, and susceptibility to landslides were successfully identified in Medellín. These locations can be assessed by the municipality of Medellín to implement LEWSs based on available budget. Implementing LEWSs in these areas has the potential to enhance the resilience of thousands of individuals residing in various parts of the city. Moreover, by utilizing the

developed cost function the price to instrumentalize the monitoring component of a LEWS in each location was estimated, and several cost-effectiveness scenarios that align with the financial resources allocated for risk management were suggested. As a result, this study provides valuable decision-making support on where to proceed with LEWS implementation following the successful deployment in Bello Oriente (Werthmann et al., 2023). With this, a conscious, informed, and transparent policy decision can be supported - where to install a LEWS under limited available financial funds. At the same time, however, the

developed scenarios show the complexity of planning and political decisions: If one decides for the most cost-effective way or to protect the most people, the most endangered areas are not necessarily instrumented. Moreover, decisions may mean reducing or increasing inequalities, depending on whether precarious settlements are preferred or not. Every decision implies advantages and disadvantages - science can at least support the choice in an information-driven way. The planning or political decision itself, however, must be fought out according to democratic processes.

This study successfully demonstrated the feasibility of accurately mapping landslide susceptibility by using a data-driven approach that integrates remotely sensed data and ancillary datasets. The application of a non-parametric random forest model allowed for the incorporation of multiple conditioning factors, capturing non-linear relationships within the data. This approach overcame the limitation posed by the non-normal distribution of factors and the requirement for prior knowledge of complex interactions among them. Consequently, an updated landslide susceptibility map was generated, complementing the city's

official hazard map from 2014. The new map incorporated new instances of mass movement obtained from landslide inventories and remote sensing.

A novel contribution was done by transparently estimating the cost of instrumentalizing the monitoring component of LEWSs, which, to the best of the authors' knowledge, has not been previously reported. This new information is valuable for decision-makers involved in disaster risk reduction, where LEWSs play a crucial role (UN, 2015). The proposed automatic monitoring

system was designed to be highly modular, scalable, and customizable, aligning with the objectives of community-based LEWSs (Gumiran et al., 2019). This design approach facilitated the transferability study, allowing for area-wide implementation across the entire city based on the experiences gained from installing a LEWS in one neighbourhood. Moreover, the system has the potential to be easily transferred to other regions worldwide with similar characteristics such as densely populated and mountainous areas. Furthermore, being based on the open LoRa® standard, the system can leverage

existing infrastructure (i.e., gateways with a cost of €2,200 each). This feature streamlines the transferability within a city and reduces costs when scaling up the systems.

The impact of various cost-effectiveness scenarios on the overall cost and the number of exposed and highly vulnerable populations when allocating a specific budget, highlighting the potential for optimization was demonstrated. The combination of a highly detailed on-site warning system with city-wide information about susceptibility and population provides unique

opportunities for researchers and decision-makers. From an economic standpoint, it is possible to determine the most effective allocation of resources in terms of the cost-to-exposed inhabitants ratio. The analysis offered a clear indication of where a

LEWS can be the most cost-effective. However, it is important to acknowledge that there are several additional factors that contribute significant value to the benefits but are difficult to quantify. These factors include reducing risks by increasing risk awareness, fostering social networks, and facilitating communication between informal dwellers and city planning agencies.

LEWSs have long being recognized for their effectiveness in providing timely warnings in at-risk communities. However, their widespread adoption has been hindered by high implementation costs. This study tackles this challenge by proposing a low-cost LEWS tailored for highly exposed areas, thus lowering the barriers to global adoption. To enhance affordability and sustainability, a wireless network of low-cost sensors equipped with solar panels is proposed. This not only improves system maintenance but also optimizes cost-effectiveness by strategically targeting areas with the highest vulnerability. By focusing 525 on these areas, the LEWS can maximize its benefits while keeping implementation costs manageable. This approach ensures that resources are allocated where they are most needed, benefiting the local community and enhancing the overall effectiveness of the system. By combining innovative technology and cost optimization strategies, this study aims to promote the wider adoption of LEWS in landslide-prone regions. By doing so, it lowers the barriers associated with expensive implementation, ultimately making LEWS more accessible and beneficial to communities at risk.

However, there are still challenges and limitations to address. In terms of the landslide susceptibility map, the accuracy was affected by evaluating the probability map using a dichotomous variable (high-hazard or non-hazard). The generalization into a dichotomous value influenced the measured overall accuracy, especially considering the frequency of values ranging between 0.4 and 0.6, which were classified as either high-hazard or non-hazard. Most of the misclassifications in the accuracy analysis occurred in the medium hazard class, which is here considered non-hazard. Additionally, misclassifications were observed in 535 low hazard areas with medium-high susceptibility values, as well as high-hazard areas with low susceptibility values. These discrepancies may arise from using two different sources as reference data for model training, which can lead to contradictions, particularly when community reports in the DAGRD inventory may not always pertain to landslides. For instance, areas with medium or low hazard levels in the POT2014 hazard map, but with similar conditioning factors to areas with recent landslides in the inventories, could exhibit high susceptibility levels in the model, resulting in validation errors. It is essential to highlight 540 that data-driven methods rely on the quality and veracity of the reference data, as well as on the conceptual approach. In this study, on the one hand, most of the landslides from inventories are reported by people, leading to a potential bias of higher landslide frequency in urban areas. However, since the focus is on implementing LEWS in urban areas, this bias is not critical to the study. Furthermore, all historic landslides in Medellin have a strong anthropic triggering factor, so it is logical to observe a higher density of landslide events in the urban areas. On the other hand, while the official hazard map is expected to be of 545 high quality due to expert evaluation, a significant number of reported landslides between 2014 and 2021 occurred in – according to the hazard map of 2014 – low hazard areas. These areas potentially include mass movements reported by citizens that may not necessarily be landslides. Another factor that may impact the results is the population increase along the urban-rural border since the creation of the official hazard map. Given the significant influence of anthropogenic factors on landslides

in Medellín, this may account for the differences between the generated landslide susceptibility map and the official landslide hazard map.

The selection of suitable exposed locations for the installation of LEWSs followed an iterative process using seeds of high susceptibly and exposure, along with thresholds set manually. These thresholds were established to identify the areas with the highest exposure based on population density and susceptibility, while also limiting the number of initial seeds to a manageable quantity. However, it is important to acknowledge that the specific thresholds and the chosen model for mapping landslide susceptibility may have influenced the identification of exposed suitable sites in Medellín. Additionally, the size of the sites was constrained based on the previous experience from Bello Oriente. These factors could impact the size, complexity, and overall accuracy of the identified sites, consequently affecting the cost estimation. Nevertheless, the final selection of more than thirty locations covers a significant portion of highly-populated exposed locations in Medellín, encompassing a diverse range of sizes and complexities. This allows for comparability in cost estimation across different sites. It is important to note that the proposed thresholds and sizes are adapted to the specific context of Medellín and should be adjusted accordingly in different regions.

The cost estimation for the proposed system is inherently subject to uncertainties, both in the cost function itself and the underlying data. An important aspect to consider is that the proposed system has only been implemented once in Bello Oriente, which possesses unique characteristics such as being an informal settlement with low-quality building structures, medium building density, steep slopes, and ample open spaces. The cost and density of sensors are based solely on the experiences at this specific site. While the cost of instruments and working hours have been provided for this case, extrapolating these costs to other areas involves certain assumptions and may not account hidden costs that could arise in different conditions. However, this cannot be assessed with the current knowledge. Additionally, local circumstances and the availability of trained personnel also contribute to the overall cost. While the absolute costs might have significant uncertainties, the proposed cost function allows for the evaluation of relative cost differences between multiple sites. Therefore, the proposed approach can be used to identify prioritized areas within a city for the initial installation of new LEWSs based on cost-effectiveness considerations.

The implementation of the LEWS prototype in Bello Oriente was a collaborative effort involving academia, private companies, government agencies, local civil society organizations, and the local community. The active involvement of these stakeholders played a vital role in addressing a range of challenges, including social conflict, insufficient risk awareness, limited political commitment, changes of local government, resources constraints, and inadequate territorial planning (Werthmann, 2023). This means that the implementation of an LEWS must always be seen as a socio-technical challenge. This involves costs that were not considered in the technical estimates.

Given the complexity of implementing a LEWS, it is important to emphasise that the estimated costs presented in this study solely pertain to the installation of the monitoring instruments. Therefore, expenses related to system maintenance, including significant working hours, additional instruments, and protective measures for the instruments, have not been accounted for.

The costs of warning elements, safety signs for emergency routes and meeting points, community engagement, and collaboration with various sectors (i.e. government, local civil society organizations, etc.), as well as the operational aspect of the LEWS, have also been excluded from the results. Finally, it is crucial to emphasize that beyond financial aspects, efficient implementation and maintenance, and cooperation among diverse stockholders, the success of the LEWS relies on social work and community engagement. The cultivation of risk awareness and fostering trust in the system define the willingness of individuals at-risk to actively participate in the process and respond promptly to warning. This social aspect is essential for ensuring the long-term sustainability of the LEWS after the monitoring instruments have been installed.

## 5 Conclusions

The implementation of multi-hazard EWSs is crucial in mitigating disaster risks and safeguarding lives, as emphasized by the Sendai Framework for Disaster Risk Reduction 2015-2030 (UN, 2015). This is particularly significant in countries like Colombia, where a considerable proportion of the population is exposed to landslide hazards and high vulnerability prevails.

Building upon the insights gained from the LEWS installed in Medellín through the Inform@risk project, this study identified 32 highly exposed areas in the city suitable for the installation of a highly modular, scalable, and customizable LEWSs. By estimating the required investment for the monitoring component of these systems, the city would need approximately €5 to €41 per protected inhabitant, varying based on site-specific characteristics.

In an approach for prioritizing the selection of exposed sites for a LEWS considering various cost-effective scenarios, budget constraints, landslide susceptibility, total population exposed, and territorial planning agenda, an informed decision-making is supported by science. These findings are intended to provide guidance to decision-makers and support disaster risk reduction efforts not only in Medellín, Colombia, but also in other regions facing similar challenges. By leveraging the lessons learned from this study, policy-makers and stakeholders can make informed decisions to enhance resilience and reduce the impact of landslides, contributing to the broader goals of disaster risk reduction.

## Data availability

All produced data can be provided by the corresponding author upon request.

## Author contribution.

Conceptualization: MS and HT; data curation: MS and MK; formal analysis: MS, MK and MG; methodology: MS; landslide susceptibility modelling: MS; cost function development: MG and JS; writing – original draft: MS; writing – review and editing: all; supervision: HT and JS; funding acquisition: HT, CGL and JS. All authors have read and agreed to the published version of the manuscript.

**Competing interests.**

The data management and analysis platform (AlpGeorisk ONLINE) used in the landslide early warning system in Bello Oriente is developed and commercially distributed by AlpGeorisk (Dr. John Singer). The other authors declare that they have no conflict of interest.

**Acknowledgements**

This research is part of the Inform@Risk project (Strengthening the resilience of informal settlements against slope
movements). We thank Leibniz Universität Hannover, Technische Hochschule Deggendorf, Technische Universität München, Deutsches Zentrum für Luft- und Raumfahrt e.V., AlpGeorisk, Sachverständigenbüro für Luftbildauswertung und Umweltfragen, EAFIT University–URBAM, Alcaldía de Medellín, Área Metropolitana del Valle de Aburrá, Sistema de Alerta Temprana del Valle de Aburrá, Sociedad Colombiana de Geología, Colectivo Tejearañas, Corporación Convivamos, Fundación Palomá and Red Barrial Bello Oriente for their support.

**Financial support**

This research was funded by the German Federal Ministry of Education and Research as part of the FONA Client II initiative, grant number 03G0883A-F. The funders had no role in study design, data collection and analysis, decision to publish, or preparation of the manuscript.

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
