# Peer review of "Cost estimation for the monitoring instrumentalization of Landslide Early Warning Systems"

_Natural Hazards and Earth System Sciences, 2023_

## Author Comment (AC1)

**Review: "Cost estimation for the monitoring instrumentalization of Landslide Early Warning Systems" by Sapena et al.**

**RC1: 'Comment on nhess-2023-41', Anonymous Referee #1**

The manuscript entitled "Cost estimation for the monitoring instrumentalization of Landslide Early Warning Systems" develops a cost-effective method for low-cost and easy-to-use EWS instrumentalization in landslide-prone areas identified based on data-driven methods. In general, the manuscript contains an interesting topic that is considered one of the important stages in landslide mechanism assessment; but there are several modifications that have to be considered. In this regard, the following comments are requested to be addressed by the authors:

We greatly appreciate the positive feedback. We would like to express our gratitude for your valuable comments and suggestions. We acknowledge the importance of addressing the modifications highlighted and assure you that we will incorporate the feedback into our work. Please find other comment-by-comment feedback in detail below.

**Comment 1:** The English of the paper is readable; however, I would suggest the authors have it checked, preferably by a native English-speaking person, to avoid any mistakes.

**Answer 1:** We agree with the reviewer that the English can be improved. We will proceed as suggested and carefully check and improve the readability of the English style by an English-speaking person.

**Comment 2:** The necessity & novelty of the manuscript should be presented and stressed in the "Introduction" section.

**Answer 2:** Thank you for highlighting this. Indeed, we believe that emphasizing the need and novelty of our study will contribute to the enhancement of our manuscript.

We will emphasize that landslide early warning systems (LEWSs) play a vital role in reducing the risk associated with landslides by providing timely information, enabling proactive measures, enhancing public awareness and education, and facilitating better planning and decision-making. Moreover, LEWSs collect data that can be used for scientific research, monitoring, and analyzing landslide behavior. This improves the understanding of landslides, their triggers, and their impacts, leading to better predictive models and more effective LEWSs [1-3].

LEWSs also provide significant economic benefits by reducing damage and loss, facilitating cost-effective planning and response, preserving economic activities, and saving costs in emergency response operations [4]. These economic advantages make investing in early warning systems (EWSs) a prudent choice. However, the cost of implementing a LEWS can vary significantly depending on various factors such as the size and complexity of the area to be monitored, the technology and infrastructure used, and the level of sophistication of the system.

As of today, estimating the costs of implementing LEWSs remains challenging due to various factors influencing the overall expenses. The implementation of a comprehensive LEWS entails substantial investments, encompassing equipment costs, infrastructure development, ongoing maintenance, and personnel expenses. Recognizing this knowledge gap, this study takes a significant leap forward by providing cost estimations for the instrumentalization of a low-cost, local, and site-specific LEWS. Furthermore, we identify highly exposed landslide-prone areas that are candidates for the installation of LEWSs in the city of Medellín. To assist decision-makers in their prioritization efforts, we offer valuable recommendations based on different cost-effectiveness scenarios. Our ultimate goal is to provide decision-makers with a comprehensive assessment that facilitates the strategic implementation of LEWSs, ensuring maximum impact and cost-efficiency.

[1] Guzzetti, F., Gariano, S. L., Peruccacci, S., Brunetti, M. T., Marchesini, I., Rossi, M., & Melillo, M. (2020). Geographical landslide early warning systems. Earth-Science Reviews, 200, 102973.

[2] WMO (2018). Multi-Hazard Early Warning Systems: A Checklist. Report. url: https://library.wmo.int/doc_num.php?explnum_id=4463. Accessed 24/05/2023.

[3] Segoni, S., Serengil, Y. & Aydin, F. A prototype landslide early warning system in Rize (Turkey): analyzing recent impacts to design a safer future. Landslides20, 683–694 (2023). Doi:10.1007/s10346-022-01988-3

[4] Rogers, D., Tsirkunov, V. (2011). Costs and benefits of early warning systems. Global assessment rep.

**Comment 3:** Provide a literature of the methods developed/applied on landslide mechanism assessment and modeling in "Introduction". The use of a table to demonstrate the advantage-disadvantage of these methods can be useful. Towards the end, mention the superiority & repeat the novelty of your work.

**Answer 3:** Thank you for your comment. In this paper, our focus was not on comparing various methods for mapping landslide susceptibility, as extensive research has already been conducted in this area [5-11]. Instead, we chose to implement a robust and validated method, specifically random forest, due to its proven accuracy in mapping landslide susceptibility, especially in Colombia when compared to other methods [7]. However, we acknowledge the importance of providing a more comprehensive overview of the existing methods in the introduction. Consequently, we will enhance the paragraph concerning the introduction of data-driven methods for mapping landslide susceptibility using EO data.

We propose to enhance the introduction with a paragraph such as:

"*Landslide susceptibility modeling has witnessed an increase in popularity due to the advancements in remote sensing, and statistical and machine learning models. Data-driven approaches have demonstrated significant potential in effectively mapping areas prone to landslides, particularly in situations where the availability of comprehensive geotechnical data required for physically-based methods are lacking. Some of the most common data-driven methods include Random forest [6-9], Logistic regression [8], Convolutional and Artificial Neuronal Networks [5,7,8], Gradient Boosted Regression Trees [7], Weight of Evidence [7], Supported Vector Machine [5,8], K-Nearest Neighbor [5,9], Naïve Bayes [9], and Linear discriminant analysis [11]. It is worth noting that, currently, there is no definitive method established as the optimal choice for empirical susceptibility modeling. In recent literature, various methods have been employed, compared, and their accuracies and suitability have shown regional variations. In this study, we implement the Random Forest method due to its demonstrated high accuracy in Colombia [7]. Additionally, this method offers the advantage of being non-parametric, allowing for the inclusion of non-normalized conditioning factors, and it is relatively straightforward to implement*".

[5] Nikoobakht, S.; Azarafza, M.; Akgün, H.; Derakhshani, R. Landslide Susceptibility Assessment by Using Convolutional Neural Network. Appl. Sci. 2022, 12, 5992. https://doi.org/10.3390/app12125992

[6] Taalab, K., Cheng, T., Zhang, Y. (2018) Mapping landslide susceptibility and types using Random Forest, Big Earth Data, 2:2, 159-178, Doi: 10.1080/20964471.2018.1472392

[7] Calderón-Guevara, W., Sánchez-Silva, M., Nitescu, B. et al. Comparative review of data-driven landslide susceptibility models: case study in the Eastern Andes mountain range of Colombia. Nat Hazards 113, 1105–1132 (2022). Doi: 10.1007/s11069-022-05339-2

[8] Ado, M.; Amitab, K.; Maji, A.K.; Jasi´nska, E.; Gono, R.; Leonowicz, Z.; Jasi´nski, M. Landslide Susceptibility Mapping Using Machine Learning: A Literature Survey. Remote Sens. 2022, 14, 3029. Doi: 10.3390/rs14133029

[9] Abu El-Magd, S.A., Ali, S.A. & Pham, Q.B. Spatial modeling and susceptibility zonation of landslides using random forest, naïve bayes and K-nearest neighbor in a complicated terrain. Earth Sci Inform 14, 1227–1243 (2021). Doi:10.1007/s12145-021-00653-y

[10] Azarafza, M., Azarafza, M., Akgün, H., Atkinson, P. M., and Derakhshani, R.: Deep learning-based landslide susceptibility mapping, Sci Rep, 11, 24112, Doi:10.1038/s41598-021-03585-1, 2021.

[11] Eiras, C. G. S., Souza, J. R. G. de, Freitas, R. D. A. de, Barella, C. F., and Pereira, T. M.: Discriminant analysis as an efficient method for landslide susceptibility assessment in cities with the scarcity of predisposition data, Nat Hazards, 107, 1427–1442, https://doi.org/10.1007/s11069-021-04638-4, 2021.

**Comment 4:** Please add a subsection clearly articulating the main limitations, wider applicability of your methods, and findings in the "Discussion" section.

**Answer 4**: Thank you for your suggestion. We appreciate your feedback regarding the discussion section of our study. In the current version of the manuscript, we focused on discussing the challenges associated with training a random forest model using different data sources and how it impacted the accuracy of mapping landslide susceptibility. We also mentioned the selection of suitable exposed sites and the need for adaptation in different regions. Additionally, we reflected on the uncertainties surrounding the cost function and the exclusion of certain costs related to maintenance, warning elements, safety signs, social work, and operating the LEWSs.

However, we recognize the importance of addressing the limitations more explicitly. In the revised version of the manuscript, we will provide a more detailed discussion of the limitations. For instance, we will delve into how the specific model applied for mapping landslide susceptibility, along with the assigned thresholds, may have influenced the identification of exposed suitable sites in Medellín. We acknowledge that these factors could impact the size, complexity, and overall accuracy of the identified sites, which in turn affect the cost estimation. Nevertheless, we would like to emphasize that our selection of 32 sites covers a significant portion of highly-populated exposed locations in Medellín, encompassing a diverse range of sizes and complexities. This allows for comparability in cost estimation across different sites.

Furthermore, we would like to acknowledge in the discussion that our cost estimation was based solely on a previous LEWS implemented in a specific neighborhood of Medellín, which exhibits certain unique characteristics such as being an informal settlement, highly vulnerable due to its low-quality building structures, with medium building density, steep slopes, and ample open spaces. We provided the cost of instruments and working hours for this specific case and used it as a basis to estimate costs in other areas, aiming to extrapolate the costs to other neighborhoods. However, we agree that it is crucial to clearly state this limitation in the manuscript, e.g. that different conditions than in our sample case may hold some hidden costs that cannot be assessed with current knowledge.

Furthermore, we recognize the importance of clearly stating the applicability of our workflow to improve the impact of our study. Therefore, we intend to incorporate a new paragraph in the discussion that highlights how our approach can be applied to other regions. For example:

"*In recent years, there has been a significant increase in the utilization of data-driven methods and EO-derived data for mapping landslide susceptibility [5-11] and generating finer-grained population distribution maps [12-13]. These advancements make it possible to identify highly exposed and landslide-prone areas in different regions worldwide. In our study, we proposed a workflow that can be applied to identify exposed sites suitable for implementing low-cost LEWSs and developed a function to estimate the cost of instrumentalization based on area, susceptibility, and building density. This approach allows for the assessment and comparison of estimated costs across multiple sites. We believe that this open and transparent cost estimation for LEWS is one of the key contributions of our study, serving as a valuable reference for other regions.*"

Regarding the findings of our study, we recognize the need to provide a concise paragraph that summarizes the key outcomes beyond what has already been discussed in the initial part of the discussion. For instance:

*"Through the application of our proposed workflow, we successfully identified critical locations characterized by high exposure, high vulnerability, and susceptibility to landslides. These locations can be assessed by the municipality of Medellín to implement LEWSs based on their available budget. Implementing LEWSs in these areas has the potential to enhance the resilience of thousands of individuals residing in various parts of the city. Moreover, by utilizing the developed cost function, we were able to suggest cost-effectiveness scenarios that align with the financial resources allocated for risk management. As a result, our study provides valuable decision-making support on where to proceed with LEWS implementation following the successful deployment in Bello Oriente [14]."*

By highlighting the significance of our findings in identifying critical locations, considering cost-effectiveness, and supporting decision-making, we aim to effectively communicate the practical implications and contributions of our study.

[12] Metzger, N., Vargas-Muñoz, J.E., Daudt, R.C. et al. (2022) Fine-grained population mapping from coarse census counts and open geodata. Sci Rep 12, 20085. Doi:10.1038/s41598-022-24495-w

[13] Sapena, M., Kühnl, M., Wurm, M., Patino, J. E., Duque, J. C., and Taubenböck, H. (2022). Empiric recommendations for population disaggregation under different data scenarios, PLoS ONE, 17, e0274504, Doi:10.1371/journal.pone.0274504.

[14] Werthmann, C., Sapena, M., Kühnl, M., Singer, J., Garcia, C., Menschik, B., Schäfer, H., Schröck, S., Seiler, L., Thuro, K., and Taubenböck, H.: Inform@Risk. The Development of a Prototype for an Integrated Landslide Early Warning System in an Informal Settlement: the Case of Bello Oriente in Medellín, Colombia, Nat. Hazards Earth Syst. Sci. Discuss. [preprint], Doi:10.5194/nhess-2023-53, in review, 2023.

**Comment 5:** The authors should deepen the discussion.

**Answer 5:** By considering the answer to the previous comment and incorporating the proposed changes, alongside our willingness to address any concerns together with the changes we plan to make based on RC2 comments, we believe that this enhances and deepens the discussion.

**Comment 6:** As a suggestion, the following articles could be useful for improving this manuscript.

1. Nikoobakht, S., Azarafza, M., Akgün, H., & Derakhshani, R. (2022). Landslide susceptibility assessment by using convolutional neural network. Applied Sciences, 12(12), 5992. https://doi.org/10.3390/app12125992

2. Fathani, T.F., Karnawati, D., Wilopo, W., Setiawan, H. (2023). Strengthening the Resilience by Implementing a Standard for Landslide Early Warning System. In: Sassa, K., Konagai, K., Tiwari, B., Arbanas, Ž., Sassa, S. (eds) Progress in Landslide Research and Technology, Volume 1 Issue 1, 2022. Progress in Landslide Research and Technology. Springer, Cham. https://doi.org/10.1007/978-3-031-16898-7_20

3. Nanehkaran, Y. A., Licai, Z., Chengyong, J., Chen, J., Anwar, S., Azarafza, M., & Derakhshani, R. (2023). Comparative Analysis for Slope Stability by Using Machine Learning Methods. Applied Sciences, 13(3), 1555. https://doi.org/10.3390/app13031555

4. Yang, F.-Y.; Zhuo, L.; Xiao, M.-L.; Xie, H.-Q.; Liu, H.-Z.; He, J.-D. A Statistical Risk Assessment Model of the Hazard Chain Induced by Landslides and Its Application to the Baige Landslide. *Appl. Sci.* 2023, *13*, 3577. https://doi.org/10.3390/app13063577

5. Gariano, S.L., Melillo, M., Brunetti, M.T., Kumar, S., Mathiyalagan, R., Peruccacci, S. (2023). Challenges in Defining Frequentist Rainfall Thresholds to Be Implemented in a Landslide Early Warning System in India. In: Sassa, K., Konagai, K., Tiwari, B., Arbanas, Ž., Sassa, S. (eds) Progress in Landslide Research and Technology, Volume 1 Issue 1, 2022. Progress in Landslide Research and Technology. Springer, Cham. https://doi.org/10.1007/978-3-031-16898-7_27

6. Segoni, S., Serengil, Y. & Aydin, F. A prototype landslide early warning system in Rize (Turkey): analyzing recent impacts to design a safer future. *Landslides*20, 683–694 (2023). https://doi.org/10.1007/s10346-022-01988-3

7. Han, Min, et al. "An Early Warning System for Landslide Risks in Ion-Adsorption Rare Earth Mines: Based on Real-Time Monitoring of Water Level Changes in Slopes." *Minerals*13.2 (2023): 265. https://doi.org/10.3390/min13020265

Thank you very much for the suggestion of these references. As suggested, we have extended the literature review for the introduction using some of these and more references, as can be seen in the answers to the first and second comments. With it, we hope we can provide a more comprehensive view on the status quo.

We express our gratitude for your valuable input, and we assure you that all of your comments and concerns will be carefully considered and incorporated into the revised manuscript.

---

## Author Response (AR1)

**Review: "Cost estimation for the monitoring instrumentalization of Landslide Early Warning Systems" by Sapena et al.**

We thank the reviewers and the editor very much for their positive feedback, suggestions and valuable comments made. We have answered to every comment in detail and made changes in the manuscript accordingly. We provide the rebuttal letter responding to all raised points from the reviewers, a marked-up copy of the manuscript with the highlighted changes and a new unmarked version of the revised manuscript. We think and hope that after those changes the manuscript has considerably improved and reflects better the purpose and limitation of our research.

**Editor: 'Comment on nhess-2023-41'**

**Comment 1:** P2, line 37: "The majority of its population lives in high and very high landslide hazard", consider revising this sentence, possibly to precise its meaning since no metrics for "high" or "very high" landslide hazard is given at this point of the manuscript.

**Answer 1:** Thank you for this observation. We agree that the sentence should be modified since the high/very high categorization of landslide hazard has not been defined in the manuscript at this point.

We have changed it to: "*The majority of its population lives in areas that are prone to landslide hazards (Ruiz Peña et al., 2017)*".

**Comment 2:** P2, lines 38-30: "This is compounded by a higher frequency of heavy and continuous precipitation, and unplanned urban growth in high-risk areas due to land scarcity", please revise the sentence and its wording. I understand that there is evidence that support a recent increase in these factors. Is it worldwide or for the specific area in Medellín, Colombia? Please be more specific.

**Answer 2:** Thank you for the observation. With this sentence, we referred to the specific case of Colombia. It is based on a report from the World Bank.

We have rewritten the sentence to: "*In Colombia, the increasing frequency of intense and persistent rainfall, coupled with unplanned urban growth in areas prone to landslides driven by limited land availability, significantly increases the likelihood of disasters, particularly impacting the most vulnerable populations (World Bank, 2012).*"

**Comment 3:** P2, paragraph in lines 41-48: Please consider revising this paragraph, there are some terms which are non-standard and would deserve more precisions. For instance, "immediate natural hazard"..."among other things" (which ones).

**Answer 3:** Thank you for your comment. We have improved the paragraph accordingly:

Now it reads, "*Due to the occurrence of a disaster, or to mitigate the effects of an imminent natural hazard, people are occasionally compelled to evacuate their places of residence. Displacement severely disrupts people's lives, rising unemployment, interrupting education, and hindering access to essential services, ultimately leading to increased vulnerability and impoverishment. Implementing preparedness measures is crucial in mitigating risks associated with displacement. These measures enhance risk awareness among individuals at risk of displacement, empowering them to make informed decisions and comply with the warnings (UNDRR, 2021)*".

**Comment 4:** P3, line 69: "Publishing on experiences, challenges, and limitations, it is nevertheless rarely done, even less so in English." What does "even less so in English"? Please be more precise, does this sentence mean that there is more literature in other languages different than English?

**Answer 4:** We apologize if this sentence was too confusing. With it, we meant that there is a lack of literature on experiences, challenges, and limitations of LEWSs. Especially, in Latin American countries, where there are a few websites, reports and thesis in Spanish, but information in English is very rare. For example:

- Sistema de Alerta Temprana ante Huaicos: Una experiencia compartida

- [Desarrollo de prototipo sistema inteligente de alerta temprana para la prevención de desastres por remoción en masa [...]](#)
- Sistema de alerta temprana de deslizamientos para el fenómeno de movimientos en masa del sector altos de la estancia
- Diseño y construcción de un prototipo de sistema de sensores inalámbricos para alerta temprana de deslaves

Therefore, we have changed the sentence by: *"The dissemination of experiences, challenges, and limitations associated with LEWSs is not a priority and is therefore rarely done, particularly in English (e.g., Reinoso Jerez, 2013; Departamento del Quindío, 2018; Castro Bonilla, 2021)."*

**Comment 5:** P20, line 438: "We proved that it is possible to map landslide susceptibility fairly accurate based on remotely sensed...". please revise the sentence and be more precise. What does "fairly accurate"? Can you give a metric or more precise meaning?

**Answer 5:** thank you for your comment. With "fairly accurate" we were making a reference to the overall accuracy presented in the result Section 3.1. After a thorough improvement of the discussion, we have rephrased the paragraph to:

*"This study successfully demonstrated the feasibility of accurately mapping landslide susceptibility by using a data-driven approach that integrates remotely sensed data and ancillary datasets…"*

**RC1: 'Comment on nhess-2023-41', Anonymous Referee #1**

The manuscript entitled "Cost estimation for the monitoring instrumentalization of Landslide Early Warning Systems" develops a cost-effective method for low-cost and easy-to-use EWS instrumentalization in landslide-prone areas identified based on data-driven methods. In general, the manuscript contains an interesting topic that is considered one of the important stages in landslide mechanism assessment; but there are several modifications that have to be considered. In this regard, the following comments are requested to be addressed by the authors:

We greatly appreciate the positive feedback. We would like to express our gratitude for your valuable comments and suggestions. We acknowledge the importance of addressing the modifications highlighted and assure you that we incorporated the feedback into our work. Please find other comment-by-comment feedback in detail below.

**Comment 1:** The English of the paper is readable; however, I would suggest the authors have it checked, preferably by a native English-speaking person, to avoid any mistakes.

**Answer 1:** We agree with you that the English had to be improved. We proceeded as suggested and carefully checked and improved the readability of the English style.

**Comment 2:** The necessity & novelty of the manuscript should be presented and stressed in the "Introduction" section.

**Answer 2:** Thank you for highlighting this. Indeed, we believe that emphasizing the need and novelty of our study will contribute to the enhancement of our manuscript.

We emphasized that landslide early warning systems (LEWSs) play a vital role in reducing the risk associated with landslides by providing timely information, enabling proactive measures, enhancing public awareness and education, and facilitating better planning and decision-making. Moreover, LEWSs collect data that can be used for scientific research, monitoring, and analyzing landslide behavior. This improves the understanding of landslides, their triggers, and their impacts, leading to better predictive models and more effective LEWSs (Guzzetti et al., 2020; WMO, 2018; Segoni et al., 2023).

LEWSs also provide significant economic benefits by reducing damage and loss, facilitating cost-effective planning and response, preserving economic activities, and saving costs in emergency

response operations (Rogers and Tsirkunov, 2011). These economic advantages make investing in early warning systems (EWSs) a prudent choice. However, the cost of implementing a LEWS can vary significantly depending on various factors such as the size and complexity of the area to be monitored, the technology and infrastructure used, and the level of sophistication of the system.

As of today, estimating the costs of implementing LEWSs remains challenging due to various factors influencing the overall expenses. The implementation of a comprehensive LEWS entails substantial investments, encompassing equipment costs, infrastructure development, ongoing maintenance, and personnel expenses. Recognizing this knowledge gap, this study takes a significant leap forward by providing cost estimations for the instrumentalization of a low-cost, local, and site-specific LEWS. Furthermore, we identify highly exposed landslide-prone areas that are candidates for the installation of LEWSs in the city of Medellín. To assist decision-makers in their prioritization efforts, we offer valuable recommendations based on different cost-effectiveness scenarios. Our ultimate goal is to provide decision-makers with a comprehensive assessment that facilitates the strategic implementation of LEWSs, ensuring maximum impact and cost-efficiency.

Please, find the changes in the revised manuscript highlighted within the 'Track Changes' version. *(See for example, lines 50-65, 87-91, and 139-142).*

Guzzetti, F., Gariano, S. L., Peruccacci, S., Brunetti, M. T., Marchesini, I., Rossi, M., & Melillo, M. (2020). Geographical landslide early warning systems. Earth-Science Reviews, 200, 102973.

Segoni, S., Serengil, Y. & Aydin, F. (2023) A prototype landslide early warning system in Rize (Turkey): analyzing recent impacts to design a safer future. Landslides20, 683–694. Doi:10.1007/s10346-022-01988-3

Rogers, D. & Tsirkunov, V. (2011). Costs and benefits of early warning systems. In: Global Assessment Report on Disaster Risk Reduction.

WMO (2018). Multi-Hazard Early Warning Systems: A Checklist. Report. url: https://library.wmo.int/doc_num.php?explnum_id=4463. Accessed 24/05/2023.

**Comment 3:** Provide a literature of the methods developed/applied on landslide mechanism assessment and modeling in "Introduction". The use of a table to demonstrate the advantage-disadvantage of these methods can be useful. Towards the end, mention the superiority & repeat the novelty of your work.

**Answer 3:** Thank you for your comment. In this paper, our focus was not on comparing various methods for mapping landslide susceptibility, as extensive research has already been conducted in this area (see references below). Instead, we chose to implement a robust and validated method, specifically random forest, due to its proven accuracy in mapping landslide susceptibility, especially in Colombia when compared to other methods (Calderón-Guevara et al., 2022). However, we acknowledge the importance of providing a more comprehensive overview of the existing methods in the introduction. Consequently, enhanced the paragraph concerning the introduction of data-driven methods for mapping landslide susceptibility using EO data.

We have modified the introduction following the recommendations, and now it reads:

*"Landslide susceptibility modeling has witnessed an increase in popularity due to the advancements in remote sensing, and machine learning models. Traditional knowledge-driven methods, such as the multicriteria analytical hierarchy process (AHP) developed by Saaty (1980), rely on weights assigned to several landslide-influencing factors. Thus, the result depends on the experience of the user and the potential to identify factors that are important for a special case (Günther et al., 2014; Skilodimou et al., 2019). In contrast, data-driven methods rely on reference data (e.g., landslide inventories) and conditioning factors, (i.e., factors influencing landslide risks), which are used to identify their interconnected relationships and predict landslide susceptibility based on statistical models. Data-driven approaches have demonstrated significant potential in effectively mapping areas prone to landslides, particularly in situations where the availability of comprehensive geotechnical data required for physically-based methods are lacking. Some of the most common data-driven methods include*

*Random forest (Taalab et al., 2018; Calderón-Guevara et al., 2022; Ado et al., 2022; Abu El-Magd et al., 2021), Logistic regression (Ado et al., 2022; Azarafza et al., 2021), Convolutional and Artificial Neuronal Networks (Nikoobakht et al., 2022; Calderón-Guevara et al., 2022; Ado et al., 2022; Azarafza et al., 2021), Boosted Regression Trees (Calderón-Guevara et al., 2022; Pourghasemi et al., 2021), Weight of Evidence (Calderón-Guevara et al., 2022), Supported Vector Machine (Nikoobakht et al., 2022; Ado et al., 2022; Azarafza et al., 2021), K-Nearest Neighbor (Nikoobakht et al., 2022; Abu El-Magd et al., 2021), Naïve Bayes (Abu El-Magd et al., 2021; Azarafza et al., 2021), and Linear discriminant analysis (Eiras et al., 2021; Pourghasemi et al., 2021). Previous studies have compared the performance of AHP and statistical methods, and the latter was found to perform better (Erener et al., 2016; Ali et al., 2021; Vojtek et al., 2021). Nevertheless, currently, there is no definitive data-driven method established as the optimal choice for empirical landslide susceptibility modeling. In recent literature, various methods have been employed, compared, and their accuracies and suitability have shown regional variations. In this study, the Random Forest method is implemented due to its demonstrated high accuracy in Colombia (Calderón-Guevara et al., 2022). Additionally, this method offers the advantage of being non-parametric, allowing for the inclusion of not-normally distributed influencing factors (Breiman, 2001)".*

Abu El-Magd, S.A., Ali, S.A. & Pham, Q.B. Spatial modeling and susceptibility zonation of landslides using random forest, naïve bayes and K-nearest neighbor in a complicated terrain. Earth Sci Inform 14, 1227–1243 (2021). Doi:10.1007/s12145-021-00653-y

Ado, M.; Amitab, K.; Maji, A.K.; Jasinska, E.; Gono, R.; Leonowicz, Z.; Jasinski, M. Landslide Susceptibility Mapping Using Machine Learning: A Literature Survey. Remote Sens. 2022, 14, 3029. Doi: 10.3390/rs14133029

Azarafza, M., Azarafza, M., Akgün, H., Atkinson, P. M., and Derakhshani, R.: Deep learning-based landslide susceptibility mapping, Sci Rep, 11, 24112, Doi:10.1038/s41598-021-03585-1, 2021.

Calderón-Guevara, W., Sánchez-Silva, M., Nitescu, B. et al. Comparative review of data-driven landslide susceptibility models: case study in the Eastern Andes mountain range of Colombia. Nat Hazards 113, 1105–1132 (2022). Doi: 10.1007/s11069-022-05339-2

Eiras, C. G. S., Souza, J. R. G. de, Freitas, R. D. A. de, Barella, C. F., and Pereira, T. M.: Discriminant analysis as an efficient method for landslide susceptibility assessment in cities with the scarcity of predisposition data, Nat Hazards, 107, 1427–1442, https://doi.org/10.1007/s11069-021-04638-4, 2021.

Nikoobakht, S.; Azarafza, M.; Akgün, H.; Derakhshani, R. Landslide Susceptibility Assessment by Using Convolutional Neural Network. Appl. Sci. 2022, 12, 5992. https://doi.org/10.3390/ app12125992

Taalab, K., Cheng, T., Zhang, Y. (2018) Mapping landslide susceptibility and types using Random Forest, Big Earth Data, 2:2, 159-178, Doi: 10.1080/20964471.2018.1472392

**Comment 4:** Please add a subsection clearly articulating the main limitations, wider applicability of your methods, and findings in the "Discussion" section.

**Answer 4**: Thank you for your suggestion. We appreciate your feedback regarding the discussion section of our study. In the previous version of the manuscript, we focused on discussing the challenges associated with training a random forest model using different data sources and how it impacted the accuracy of mapping landslide susceptibility. We also mentioned the selection of suitable exposed sites and the need for adaptation in different regions. Additionally, we reflected on the uncertainties surrounding the cost function and the exclusion of certain costs related to maintenance, warning elements, safety signs, social work, and operating the LEWSs.

However, we recognize the importance of addressing the limitations more explicitly. In the revised version of the manuscript, we provided a more detailed discussion of the limitations. For instance, we delved into how the specific model applied for mapping landslide susceptibility, along with the assigned thresholds, may have influenced the identification of exposed suitable sites in Medellín. We acknowledge that these factors could impact the size, complexity, and overall accuracy of the identified sites, which in turn affect the cost estimation. Nevertheless, we would like to emphasize that our selection of 32 sites covers a significant portion of highly-populated exposed locations in Medellín,

encompassing a diverse range of sizes and complexities. This allows for comparability in cost estimation across different sites.

Furthermore, we acknowledged in the discussion that our cost estimation was based solely on a previous LEWS implemented in a specific neighborhood of Medellín, which exhibits certain unique characteristics such as being an informal settlement, highly vulnerable due to its low-quality building structures, with medium building density, steep slopes, and ample open spaces. We provided the cost of instruments and working hours for this specific case and used it as a basis to estimate costs in other areas, aiming to extrapolate the costs to other neighborhoods. However, we agree that it is crucial to clearly state this limitation in the manuscript, e.g. that different conditions than in our sample case may hold some hidden costs that cannot be assessed with current knowledge.

Furthermore, we recognize the importance of clearly stating the applicability of our workflow to improve the impact of our study. Therefore, we incorporated a new paragraph in the discussion that highlights how our approach can be applied to other regions:

" *In recent years, there has been a significant increase in the utilization of data-driven methods and EO-derived data for mapping landslide susceptibility (e.g., Abu El-Magd et al., 2021; Ado et al., 2022; Azarafza et al., 2021; Calderón-Guevara et al., 2022; Eiras et al., 2021; Nikoobakht et al., 2022; Taalab et al., 2018) and for generating detailed population distribution maps (Sapena et al., 2022; Metzger et al., 2022). These advancements have enabled the identification of highly exposed areas prone to landslides in various regions worldwide (Garcia et al., 2016; Modugno et al., 2022; Kühnl et al., 2022). This study proposes a comprehensive workflow that can be applied to identify exposed areas prone to landslides suitable for the implementation of low-cost and site-specific LEWSs. Furthermore, it adds to current literature as a cost estimation function for the instrumentalization of the LEWSs is developed, considering factors such as area, landslide susceptibility, and building density, allowing for the assessment and comparison of estimated costs across multiple sites. This integrated approach facilitates informed decision-making processes by prioritizing actions based on cost-effectiveness. One of the key contributions of this study is the provision of an open and transparent cost estimation for LEWSs, serving as a valuable reference for other regions.*".

Regarding the findings of our study, we recognize the need to provide a concise paragraph that summarizes the key outcomes beyond what has already been discussed in the initial part of the discussion, such as:

"*Through the application of the proposed workflow, more than thirty critical locations characterized by high exposure, high vulnerability, and susceptibility to landslides were successfully identified in Medellín. These locations can be assessed by the municipality of Medellín to implement LEWSs based on available budget. Implementing LEWSs in these areas has the potential to enhance the resilience of thousands of individuals residing in various parts of the city. Moreover, by utilizing the developed cost function the price to instrumentalize the monitoring component of a LEWS in each location was estimated, and several cost-effectiveness scenarios that align with the financial resources allocated for risk management were suggested. As a result, this study provides valuable decision-making support on where to proceed with LEWS implementation following the successful deployment in Bello Oriente (Werthmann et al., 2023). With this, a conscious, informed, and transparent policy decision can be supported - where to install a LEWS under limited available financial funds. At the same time, however, the developed scenarios show the complexity of planning and political decisions: If one decides for the most cost-effective way or to protect the most people, the most endangered areas are not necessarily instrumented. Moreover, decisions may mean reducing or increasing inequalities, depending on whether precarious settlements are preferred or not. Every decision implies advantages and disadvantages - science can at least support the choice in an information-driven way. The planning or political decision itself, however, must be fought out according to democratic processes.*"

By highlighting the significance of our findings in identifying critical locations, considering cost-effectiveness, and supporting decision-making, we aim to effectively communicate the practical implications and contributions of our study.

Metzger, N., Vargas-Muñoz, J.E., Daudt, R.C. et al. (2022) Fine-grained population mapping from coarse census counts and open geodata. Sci Rep 12, 20085. Doi:10.1038/s41598-022-24495-w

Sapena, M., Kühnl, M., Wurm, M., Patino, J. E., Duque, J. C., and Taubenböck, H. (2022). Empiric recommendations for population disaggregation under different data scenarios, PLoS ONE, 17, e0274504, Doi:10.1371/journal.pone.0274504.

Werthmann, C., Sapena, M., Kühnl, M., Singer, J., Garcia, C., Menschik, B., Schäfer, H., Schröck, S., Seiler, L., Thuro, K., and Taubenböck, H.: Inform@Risk. The Development of a Prototype for an Integrated Landslide Early Warning System in an Informal Settlement: the Case of Bello Oriente in Medellín, Colombia, Nat. Hazards Earth Syst. Sci. Discuss. [preprint], Doi:10.5194/nhess-2023-53, in review, 2023.

**Comment 5:** The authors should deepen the discussion.

**Answer 5:** By considering the answer to the previous comment and incorporating the proposed changes, alongside with the changes we made based on RC2 comments, we believe that we enhanced and deepened the discussion.

**Comment 6:** As a suggestion, the following articles could be useful for improving this manuscript.

1. Nikoobakht, S., Azarafza, M., Akgün, H., & Derakhshani, R. (2022). Landslide susceptibility assessment by using convolutional neural network. Applied Sciences, 12(12), 5992. https://doi.org/10.3390/app12125992

2. Fathani, T.F., Karnawati, D., Wilopo, W., Setiawan, H. (2023). Strengthening the Resilience by Implementing a Standard for Landslide Early Warning System. In: Sassa, K., Konagai, K., Tiwari, B., Arbanas, Ž., Sassa, S. (eds) Progress in Landslide Research and Technology, Volume 1 Issue 1, 2022. Progress in Landslide Research and Technology. Springer, Cham. https://doi.org/10.1007/978-3-031-16898-7_20

3. Nanehkaran, Y. A., Licai, Z., Chengyong, J., Chen, J., Anwar, S., Azarafza, M., & Derakhshani, R. (2023). Comparative Analysis for Slope Stability by Using Machine Learning Methods. Applied Sciences, 13(3), 1555. https://doi.org/10.3390/app13031555

4. Yang, F.-Y.; Zhuo, L.; Xiao, M.-L.; Xie, H.-Q.; Liu, H.-Z.; He, J.-D. A Statistical Risk Assessment Model of the Hazard Chain Induced by Landslides and Its Application to the Baige Landslide. *Appl. Sci.* 2023, *13*, 3577. https://doi.org/10.3390/app13063577

5. Gariano, S.L., Melillo, M., Brunetti, M.T., Kumar, S., Mathiyalagan, R., Peruccacci, S. (2023). Challenges in Defining Frequentist Rainfall Thresholds to Be Implemented in a Landslide Early Warning System in India. In: Sassa, K., Konagai, K., Tiwari, B., Arbanas, Ž., Sassa, S. (eds) Progress in Landslide Research and Technology, Volume 1 Issue 1, 2022. Progress in Landslide Research and Technology. Springer, Cham. https://doi.org/10.1007/978-3-031-16898-7_27

6. Segoni, S., Serengil, Y. & Aydin, F. A prototype landslide early warning system in Rize (Turkey): analyzing recent impacts to design a safer future. *Landslides* 20, 683–694 (2023). https://doi.org/10.1007/s10346-022-01988-3

7. Han, Min, et al. "An Early Warning System for Landslide Risks in Ion-Adsorption Rare Earth Mines: Based on Real-Time Monitoring of Water Level Changes in Slopes." *Minerals* 13.2 (2023): 265. https://doi.org/10.3390/min13020265

Thank you very much for the suggestion of these references. As suggested, we have extended the literature review for the introduction using some of these and more references, as can be seen in the answers to the first and second comments. With it, we hope we can provide a more comprehensive view on the status quo.

We express our gratitude for your valuable input, and we assure you that all of your comments and concerns were carefully considered and incorporated into the revised manuscript.

Technological advances in recent years, and in particular the emergence of the Internet of Things paradigm, low-cost programmable electronics and sensors, ubiquitous (wireless) communication infrastructure and a wide range of data-based services and resources available on the Internet, bear the great potential for filling a much-needed gap of monitoring high-risk geographic areas with sufficient density in order to allow for effective early warning of disasters of natural origin. The potential merits of such technological solutions are nevertheless challenged by still hard to quantify costs and benefits, therefore slowing down their adoption by decision makers. The paper deals with this relevant topic and offers some new insights into the cost of implementing an early warning system for landslides. The work is based on data gathered from field work, experience with a real-life sensor network deployment, and analysis of various databases of historical landslide events. This is a clear strength of the work.

We sincerely appreciate the positive feedback. The recognition of the paper's relevance and contribution to providing new insights into the cost of implementing a LEWS is encouraging. We are grateful for the acknowledgement of the work's strength. We would like to assure the reviewer that we carefully addressed all concerns and incorporated the suggestions into an improved version of the manuscript. We were committed to enhancing the quality of our manuscript based on the reviewers' comments.

Despite the above, I have three main concerns regarding the article in its current form:

**Concern no. 1:** Section 2 on material and methods provides a description of a number of tasks undertaken for merging and filtering databases and geographic information systems about historic landslides, field observation and demographic information in order to identify the most suitable candidate areas for deploying monitoring sensor networks. Regretfully, however, the description lacks enough detail for the work to be reproducible. To give a few examples on p. 8: Statements such as "...we use topographic, geological and precipitation factors..." (What are these factors? How are they used?), "...we rely on socio-demographic factors..." (same as before), "...we tested several methods..." (Which methods?) "...and 500 pixels proved to be the most appropriate..." (By what criterion?), are not sufficient in a scientific article.

**Answer 1:** Thank you very much for your comment. We agree that section 2.3 lacks certain information to make the work completely reproducible. We apologize for that, we shortened this section in the original manuscript for the brevity of the manuscript. Based on your suggestion, we adapted this part now to be clearer.

Regarding the first paragraph, where we mentioned without any further detail the topographic, geological, precipitation and socio-demographic factors, we did it in an introductory sense for section 2.3. These factors and their preprocessing are explained in the successive paragraphs (i.e., topography-derived factors new lines 211-230; geological-derived factors new lines 231-235; precipitation-derived factor new lines 236-246; and socio-economic factors new lines 247-256). However, we understand that this can be misleading. Therefore, we improved the introductory line as follow:

"*For modelling landslide susceptibility, the proposed methodology incorporates a range of factors influencing landslide risks including topographic, geological, and precipitation data. In addition, to support the search for suitable locations for the implementation of LEWSs, socio-demographic factors are also considered. The database consists of the official cartography from open data platforms of the city of Medellín and the metropolitan area of the Aburrá valley such as 'GeoMedellín' and 'Datos Abiertos del AMVA' (Alcaldía de Medellín, 2023; AMVA, 2023), precipitation data from SIATA (SIATA, 2023), OpenStreetMap data (downloaded in 2022, openstreetmap.org), a high resolution population map that provides estimates of the number of people per building and a grid of 100 meters from Sapena et al. (2022), and a map of precarious settlements from Kühnl et al. (2021)*".

With regard to the flow accumulation threshold for mapping the stream network, the selection of an appropriate value is influenced by the desired stream density and the specific characteristics of the

study area. While a universally agreed threshold value does not exist, it is common to use values ranging from approximately 100 to 1000 pixels or 0.05 to 5 km$^2$ drainage area, considering factors such as data scale, resolution, and landscape attributes [15, 16]. In our study, we tested several values across this range and found that 500 contributing pixels (i.e., 12,500 m²) yielded the closest correspondence with the official drainage system map from the Master plan (POT) of Medellín. Furthermore, a visual evaluation of the results revealed a satisfactory representation of the majority of streams. It is worth emphasizing that the threshold value is subject to variation based on the study requirements and the user's preferences. We acknowledge the importance of highlighting this aspect in the new version of the manuscript. Therefore, we clarified the paragraph as follows:

*"Regarding the stream network, there is no universally agreed upon flow accumulation threshold for determining streams due to its dependence on various factors, including desired stream density, data scale, resolution, and landscape attributes. Nonetheless, it is common practice to employ threshold values within a range of approximately 100 to 1000 pixels or 0.05 to 5 km2 drainage area (Reddy et al., 2018; Tarboton et al., 1991). In this study, multiple values within this range are evaluated, and determined that a threshold of 500 contributing pixels (equivalent to a minimum stream inflow of 12,500 m²) achieved the closest correspondence with the official drainage system map from the POT of Medellín. Furthermore, a visual examination of the result reveals a satisfactory representation of the majority of streams."*

Reddy, G. O., Kumar, N., Sahu, N., & Singh, S. K. (2018). Evaluation of automatic drainage extraction thresholds using ASTER GDEM and Cartosat-1 DEM: A case study from basaltic terrain of Central India. The Egyptian Journal of Remote Sensing and Space Science, 21(1), 95-104.

Tarboton, D. G., Bras, R. L., and Rodriguez-Iturbe, I.: On the extraction of channel networks from digital elevation data, Hydrol. Process., 5, 81–100, https://doi.org/10.1002/hyp.3360050107, 1991.

**Concern no. 2.:** For the cost evaluation the authors emphasize that only the costs of the implementation of the wireless geosensor network are included in the cost function, while cost of aspects such as risk evaluation, social interventions, social work and network maintenance are not included. While it is valid to isolate the cost of the technology (the geosensor network) form other costs, it is also the case that a number of cost factors considered for the technology in this work are missing. In particular, the cost of operating the network over time is an essential aspect that decision-makers must know. Securing budget for just deploying the network is not enough: it's wasted money if not accompanied by proper maintenance. Maintenance costs include replacement parts, the cost of vandalized or stolen sensor nodes, cost of human resources in the field for doing maintenance and in the lab for repairs, plus tools for fieldwork, transport to and from sites, etc. Other operational costs include the cost of Internet access to and from the gateways, cost of databases and severs in the Cloud and possibly software tools. The cost of manufacturing the sensor nodes should include a yield factor for the various parts (purchased off the shelf or manufactured in house). Finally, a realistic cost estimate should include some rental cost for office and workshop space and utilities (electricity at least). Otherwise, who pays for that once the decision-maker places an order?

**Answer 2:** Thank you for your comment. We understand your concern and we acknowledge the importance of obtaining accurate figures regarding the operational and maintenance costs of LEWSs. However, it is important to note that the LEWS currently installed in Bello Oriente, that we used as a reference in the cost estimation for the instrumentalization, is a prototype (fully installed) and not yet fully operational. As a result, we lack the necessary experience and data to provide precise estimates for the "normal" operation costs of a LEWS. While we could speculate on potential numbers, we are cautious about making inaccurate guesses that lack a solid foundation for publication. We believe that it would be more reliable to wait until the system has been operational for at least one year, as this would allow us to gather sufficient data to provide meaningful cost estimates.

**Concern no. 3:** The argumentation about cost-effectiveness in Section 3 centers on various ways of prioritizing how to choose the deployment sites under budget restrictions. While considering these aspects does make sense, there is a more fundamental prior question that is not addressed nor

discussed: what is the benefit of deploying the technology at each site? What is the cost-benefit relationship for each site, is it worth the cost? Intuition says it is, but only if the answer is yes, then the presented prioritization comes into play. Similarly, the argument on p. 18 "if the city would use the same budget to instrumentalize 9 EWS" is hard to follow: the city probably does prioritize based on cost-benefit and (correctly) decides to prioritize other expenses. The argument about re-balancing budget only holds water if the cost-benefit relationships are understood.

**Answer 3:** Thank you for raising these issues. LEWSs are widely recognized as effective means of providing timely warnings. However, the high costs associated with their implementation have hindered their global adoption. In this study, our aim is to address some of these limitations by proposing a low-cost LEWS specifically designed for highly exposed areas. To enhance the affordability and sustainability of the system, the wireless network of low-cost sensors, which have solar panels, is based on Internet of Things (IoT) technologies. This approach not only improves system maintenance but also optimizes cost-effectiveness by strategically targeting areas with the most highly vulnerable populations. By focusing on the highly vulnerable areas, we can maximize the benefits of the LEWS while keeping implementation costs manageable. This approach ensures that resources are allocated where they are most needed, benefiting the local community and enhancing the overall effectiveness of the system. By combining innovative technology, community involvement, and cost optimization strategies, our study aims to overcome the barriers associated with expensive implementation and promote the wider adoption of LEWS in landslide-prone areas.

With respect to your very important questions "what is the benefit of deploying the technology at each site? What is the cost-benefit relationship for each site, is it worth the cost?" – this is of course, very difficult to answer. From an economic point of view, we can determine where it is best employed to have the best ratio cost vs. exposed inhabitants. However, it is clear that there are more aspects with respect to saving lives, social networks, communication between informal dwellers and the city-planning agency, among many more that have a high value which is difficult to quantify.

In the supplementary material, we included Table S1 and Figure S1 to provide a comprehensive summary of this information (https://nhess.copernicus.org/preprints/nhess-2023-41/nhess-2023-41-supplement.pdf). These present a site-by-site analysis, indicating the potential locations for LEWS implementation, the population that would be protected, and the breakdown of highly vulnerable individuals within that population. Additionally, the table includes the average landslide susceptibility and other relevant factors. Moreover, we have included the estimated cost to implement the LEWS for each site, as well as the cost per person. This allows for a clear understanding of the financial implications and cost-effectiveness of the proposed LEWS implementation in each specific location, enabling decision-makers to make informed choices regarding the prioritization and implementation of LEWS.

We have taken into consideration your concerns and acknowledge the need to further address the cost-benefit relationship of deploying LEWS at each site. Section 3.3 and the supplementary material provide a foundation for understanding the benefits associated with the implementation of LEWS at different locations and their cost-effectiveness. To address these concerns more comprehensively, we delved deeper into the cost-benefit relationship in the discussion section of the paper:

"*From an economic standpoint, it is possible to determine the most effective allocation of resources in terms of the cost-to-exposed inhabitants ratio. The analysis offered a clear indication of where a LEWS can be the most cost-effective. However, it is important to acknowledge that there are several additional factors that contribute significant value to the benefits but are difficult to quantify. These factors include reducing risks by increasing risk awareness, fostering social networks, and facilitating communication between informal dwellers and city planning agencies.*"

"*LEWSs have long being recognized for their effectiveness in providing timely warnings in at-risk communities. However, their widespread adoption has been hindered by high implementation costs. This study tackles this challenge by proposing a low-cost LEWS tailored for highly exposed areas, thus lowering the barriers to global adoption. To enhance affordability and sustainability, a wireless network*"

*of low-cost sensors equipped with solar panels is proposed. This not only improves system maintenance but also optimizes cost-effectiveness by strategically targeting areas with the highest vulnerability. By focusing on these areas, the LEWS can maximize its benefits while keeping implementation costs manageable. This approach ensures that resources are allocated where they are most needed, benefiting the local community and enhancing the overall effectiveness of the system. By combining innovative technology and cost optimization strategies, this study aims to promote the wider adoption of LEWS in landslide-prone regions. By doing so, it lowers the barriers associated with expensive implementation, ultimately making LEWS more accessible and beneficial to communities at risk.."*

Regarding the argument mentioned on page 18, our intention was to highlight the potential of the introduced cost function by providing an example of how many LEWS could be implemented with a specific budget (i.e., COP 2,000,000,000 (≈ €397,000)), considering different cost-effectiveness scenarios. The purpose of this demonstration was to illustrate how different site selection strategies, based on various priority scenarios, can impact the total number of implemented LEWS, the population covered, the cost per person, and other relevant factors. We recognize the need for clarification in this section to ensure a better understanding of the purpose behind this exemplary exercise. We improved the explanation to clearly communicate the objective, emphasizing how it serves to demonstrate the relationship between budget allocation, site prioritization strategies, and the resulting outcomes.

**Comment 4:** On a related matter, a number of potential sites are ruled out because it is complicated to deploy networks there. What is the proportion of population that could benefit from the monitoring networks if unlimited budget was available? How much population would need other kinds of solutions?

**Answer 4:** Thank you for addressing these questions. By combining the landslide susceptibility map with the population map (Figure 4), we have identified a considerable number of individuals who are potentially at risk. Specifically, we found that there are 172,721 people residing in areas with a very high landslide susceptibility level (≥ 0.7) (Figure 4B, red cells). Among these individuals, 165,247 are considered highly vulnerable (which accounts for 95% of the exposed population). Furthermore, we have determined that there is a population of 379,308 individuals living in areas exposed to medium-high landslide susceptibility (ranging between 0.5 and 0.7 susceptibility, Figure 4B, orange cells). Among this group, 301,278 people are considered highly vulnerable, which represents approximately 79% of the total population in those areas.

These findings indicate that a significant portion of Medellín's population is settled in landslide-prone areas. Specifically, 7.5% of the total population resides in very high landslide susceptibility zones, while 16.4% are settled in medium-high landslide susceptibility areas. It is noteworthy that more than half a million people live in areas with a landslide susceptibility level of at least 0.5. If financial constraints were not a limitation, the implementation of LEWSs could greatly benefit these individuals. However, prior to implementing the LEWs, it is crucial to conduct inspections in the respective areas to assess the suitability of the system in each location.

When addressing the challenges of implementing solutions in highly-dense built-up areas where subsurface sensors are not feasible, we propose the implementation of a network of LoRa sensors as an alternative approach. These sensors can provide valuable data for monitoring and detecting landslide risks in these areas. However, the information is more uncertain as a result of the absence of subsurface monitoring. To assess the feasibility of this alternative configuration, we suggest utilizing the proposed function to estimate the associated costs.

**Comment 5:** There are a number of specific comments and technical corrections that should be addressed. The following list is not comprehensive:

1. P. 2: "among many other things" is somewhat too colloquial for a research paper. "Untapped potentials", should read "untapped potential".

   Thank you for noticing, we have addressed these issues and improve the overall writing style of the paper as suggested also by RC1.

2.  P. 7: "30.200 report of potential mass movements": how many of those are unique and distinct events?

Unfortunately, the employed DAGRD landslide database has certain limitations. Firstly, the database may suffer from inaccuracies since it relies on citizen reports before technicians visit the site to classify whether an event is a mass movement or not. Consequently, incidents such as cracks or humidity could be incorrectly reported as mass movements. Secondly, the reports are limited to urban areas, which may have only a minor impact on our study since we combine susceptibility with populated areas. To mitigate these limitations, we have incorporated additional databases, including other landslide databases, remotely-sensed landslides, and the hazard map. By prioritizing the locations from these databases in the selection of training data, we aim to reduce the influence of non-mass movement reports. Additionally, when choosing sample locations, we ensure a minimum distance of 5 meters between each sampled point, corresponding to the spatial resolution of the spatial data, to avoid selecting samples within the same pixel. As explained in section 2.4, we have utilized 2,800 recorded landslide events from the combined databases and 5,000 from the hazard map, which were then split into train and test data sets. This approach is intended to minimize the impact of false reports on our analysis.

Unfortunately, we cannot determine the exact number of unique reports in the DAGRD landslide inventory. However, we believe that if two distinct locations experience structural damage due to a common landslide, even if they are geographically distant from each other, they can still contribute to training a model aimed at mapping landslide susceptibility.

3.  P. 9: "measurements from 215 stations"? Are these all stations available in the Aburrá valley, is this a subset? How were they chosen? "Root mean square error of 506 mm". Seems a lot. Can you put it in perspective, give some insight? What is a number that could be expected?

Yes. We used all available stations in the Aburrá valley, which include a total of 215 stations. Our aim was to generate a continuous map of rainfall data by interpolating the accumulated precipitation values over the course of a year from these stations. To achieve this, we employed the Ordinary Kriging Optimized Smoothed (OKOS) method, accompanied by cross-validation. For this process, we selected 70% of the stations for training and utilized the remaining 30% for testing. Our evaluation provided a Root Mean Square Error (RMSE) value of 506 mm/year, which represents the discrepancy between the predicted and observed yearly precipitation accumulation. Being the mean value of precipitation across all stations 1,425 mm/year, the normalized RMSE, indicating the relative error, is approximately ± 35%. Although this may seem like a significant error, it is consistent with existing literature, which commonly reports normalized RMSE values ranging from 30-35% [17] and 17-29% [18].

We selected the OKOS method based on its lower RMSE compared to other methods tested during the process of interpolating annual precipitation. To determine the most accurate interpolation, we conducted six test rounds using two commonly recommended methods validated in the literature [19]: Inverse Distance Weighting (IDW) with three rounds and Ordinary Kriging (OK) with three rounds, each with different settings.

*Please, for specific changes go to the new version of the manuscript lines 236-246.*

[17] Bostan, P. A., Heuvelink, G. B., Akyurek, S. Z. (2012). Comparison of regression and kriging techniques for mapping the average annual precipitation of Turkey. International Journal of Applied Earth Observation and Geoinformation, 19, 115-126.

[18] Antal, A., Guerreiro, P.M.P., Cheval, S. (2021). Comparison of spatial interpolation methods for estimating the precipitation distribution in Portugal. Theor Appl Climatol 145, 1193–1206. Doi: 10.1007/s00704-021-03675-0

[19] Ly, S., Charles, C., Degré, A. (2013). Different methods for spatial interpolation of rainfall data for operational hydrology and hydrological modeling at watershed scale. A review. Biotechnol. Agron. Soc. Environ.,17(2), 292-406.

4. P. 10: "Training a statistical model". Which one? Can you give a reference?

With "statistical model" we refer to the Random Forests model that is explained afterwards. Random forest is a statistical- or machine-learning algorithm used for prediction, in this case the probability of a landslide event. To clarify this, we modified the sentence by stating: "*to train a Random Forest statistical model*".

5. P. 11: The word "sensor" is used in place of "node" (one node can have several sensors). This confuses the reader.

Thank you for this comment. We checked the use of the words sensor and node and modified them accordingly based on the context.

6. P17.: "Amount of people". Better word: population. The word "cheap" should be avoided. In-expensive, low-cost.

Thank you for pointing out these issues. We will address them. We will replace "*amount of people*" with "*population*" and "*cheapest*" with "*least expensive*".

7. Table S1 is cited several times, but it is not available.

We apologize for the misunderstanding, we provided Table S1 and Figure S1 in the supplementary material that can be found on the Preprint nhess-2023-41 – Supplement link or on the following link: https://nhess.copernicus.org/preprints/nhess-2023-41/nhess-2023-41-supplement.pdf

8. Acronyms not defined: CSM-EXT, AOI

Thank you. We introduced the Continuous Shear Monitor measurement cable and extensometers (CSM-EXT) term, while the Area of Interest (AOI) used in the formula was changed by "*site*" in this case, since it is more appropriate.

9. All pages: phrases beginning with "we" are by far too many, this should be revised.

Thank you for your feedback. We appreciate your comment. We improved the writing style of the manuscript. We minimized the use of "we" throughout the document.

I encourage the authors to revise their paper and improve it, because this kind of work, while complex and difficult to do, is very much necessary.

We sincerely appreciate your positive feedback and the valuable insights you have provided. Your comments and concerns have been duly noted, and addressed each of them in the revised version of the manuscript.